# Hepatic stellate cell stearoyl co-A desaturase activates leukotriene B4 receptor 2 - β-catenin cascade to promote liver tumorigenesis

Sonal Sinha[1,2], Satoka Aizawa[1,2], Yasuhiro Nakano[3], Alexander Rialdi[4], Hye Yeon Choi[1,2], Rajan Shrestha[1,2], Stephanie Q. Pan[1,2], Yibu Chen[5], Meng Li[5], Audrey Kapelanski-Lamoureux[6], Gregory Yochum[7], Linda Sher[8], Satdarshan Paul Monga[9], Anthoula Lazaris[6], Keigo Machida[1,10], Michael Karin[11], Ernesto Guccione[4] & Hidekazu Tsukamoto[1,2,12] ✉

Hepatocellular carcinoma (HCC) is the 3rd most deadly malignancy. Activated hepatic stellate cells (aHSC) give rise to cancer-associated fibroblasts in HCC and are considered a potential therapeutic target. Here we report that selective ablation of stearoyl CoA desaturase-2 (Scd2) in aHSC globally suppresses nuclear CTNNB1 and YAP1 in tumors and tumor microenvironment and prevents liver tumorigenesis in male mice. Tumor suppression is associated with reduced leukotriene B4 receptor 2 (LTB4R2) and its high affinity oxylipin ligand, 12-hydroxyheptadecatrienoic acid (12-HHTrE). Genetic or pharmacological inhibition of LTB4R2 recapitulates CTNNB1 and YAP1 inactivation and tumor suppression in culture and in vivo. Single cell RNA sequencing identifies a subset of tumor-associated aHSC expressing Cyp1b1 but no other 12-HHTrE biosynthetic genes. aHSC release 12-HHTrE in a manner dependent on SCD and CYP1B1 and their conditioned medium reproduces the LTB4R2-mediated tumor-promoting effects of 12-HHTrE in HCC cells. CYP1B1-expressing aHSC are detected in proximity of LTB4R2-positive HCC cells and the growth of patient HCC organoids is blunted by LTB4R2 antagonism or knockdown. Collectively, our findings suggest aHSC-initiated 12-HHTrE-LTB4R2-CTNNB1-YAP1 pathway as a potential HCC therapeutic target.

Activated hepatic stellate cells (aHSC) promote HCC as cancer-associated fibroblasts (CAF)[1–3]. Metabolic liver diseases such as alcoholic and non-alcoholic steatohepatitis are now recognized as the dominant HCC predisposing conditions, taking the place of viral hepatitis[4,5]. However, the specific nature of aHSC-HCC crosstalk in HCC development in such conditions is yet to be elucidated. We recently showed conditional ablation of stearoyl CoA desaturase 2 (Scd2) in Col1a1-expressing cells in Scd2^{f/f} mice carrying Col1a1-Cre (Scd2^{f/f};

Col1a1-Cre) attenuated HCC development initiated by diethyl nitrosamine (DEN) and promoted by Western alcohol diet (WAD)[6]. As aHSC represents a dominant Col1a1-expressing cell type in multiple liver injury models[7], our results suggested the tumor-promoting role of Scd2 in aHSC. Molecular dissection revealed that Scd2, the isoform expressed by mouse HSC, is transcriptionally activated by β-catenin (CTNNB1) via its interaction with SREBP-1c bound to the Scd2 promoter. SCD2 establishes a positive forward loop for the Wnt pathway via ELAV1

(HUR)-mediated stabilization of *Lrp5/6* mRNA[8]. This SCD-Wnt positive loop is required for activation of HSC and self-renewal of mouse liver tumor-initiating stem cells (TIC) which also express *Scd2* as opposed to *Scd1* by hepatocytes[9], suggesting the pathway shared by aHSC and TIC may support liver tumorigenesis.

The present study shows that conditional ablation of *Scd2* in aHSC causes global inhibition of the SCD-Wnt positive loop and leukotriene B4 receptor 2 (LTB4R2)-dependent CTNNB1-YAP1 pathway, suppressing liver tumorigenesis, the effects recapitulated by LTB4R2 antagonism or knockdown.

## Results

### Global YAP1 repression by aHSC SCD2 deficiency
We first aimed to validate the use of *Scd2*[f/f]*;Col1a1Cre* (*Scd2*[f/f]*;CC*) mice for selective *Scd2* ablation in aHSC. For this, we performed single-cell RNA sequencing (scRNA-seq) of DEN + WAD mouse liver cells. This analysis identified clusters of different cell types expressing their marker genes (Supplementary Fig. 1a). Among them, *Scd2* was expressed by *Lyve1*[+] endothelial cells, *Adgre1*(*F4/80*)[+] macrophages, *Lrat*[+] HSC, *Fbln2*[+] portal fibroblasts (PF), and *Itgax* (*Cd11c*)[+] dendritic cells. However, *Col1a1* was selectively expressed by *Lrat*[+] HSC and *Fbln2*[+] PF (Supplementary Fig. 1b). Thus, *Scd2* ablation by *Col1a1* promoter-induced Cre should selectively occur in these two cell types in *Scd2*[f/f]*;CC* mice. Further, we showed ~60% of these *Fbln2*[+] cells were *Lrat*[+] aHSC (Supplementary Fig. 1c) as we recently reported for other models[10]. Collectively, these results support that *Scd2* ablation is relatively selective in aHSC in *Scd2*[f/f]*;CC* mice. In fact, *Scd2* expression in HSC isolated from *Scd2*[f/f]*;CC* vs. *Scd2*[f/f] DEN + WAD mice was 20-folder lower (Supplementary Fig. 1d), supporting the effectiveness of this genetic strategy.

Next, we unbiasedly explored the mechanisms of the anti-tumor effect of aHSC SCD2 deficiency by RNA sequencing (RNA-seq) of tissues immediately adjacent to liver tumors of *Scd2*[f/f]*; CC* vs. *Scd2*[f/f] mice. We considered these tissues referred to tumor-adjacent livers (TAL) are ideal for studying aHSC-tumor cell crosstalk as they contained microscopic tumors although visible tumors were not evident. A heatmap of differentially expressed genes (DEGs), depicted a distinct transcriptomic landscape of *Scd2*[f/f]*;CC* vs. *Scd2*[f/f] (Fig. 1a). Evaluation of DEGs (Supplementary Data 1), identified downregulation of genes relevant to pathways of interest in *Scd2*[f/f]*;CC* (Supplementary Data 2) including tumor development and HSC activation, the results expected from the *Scd2*[f/f]*;CC* mouse phenotype. Wnt-CTNNB1 pathway was also downregulated, suggesting that selective inhibition of SCD-Wnt positive loop in aHSC, globally suppressed this pathway. Indeed, immunoblotting (IB) analysis revealed that the key components of the loop (CTNNB1, HUR, LRP6), were reduced in *Scd2*[f/f]*;CC* (Fig. 1b). YAP1 protein and mRNA upregulations in *Scd2*[f/f] were also prevented in *Scd2*[f/f]*;CC* (Fig. 1b and g). TAZ also contributes to liver carcinogenesis, particularly in c-Myc-induced HCC[11]. Similar reductions in TAZ protein and *Wwtr1* mRNA were observed in *Scd2*[f/f]*;CC* (Supplementary Fig. 1e, f). To characterize YAP1 repression at the cellular level, we examined co-expression of YAP1 and cell-type markers by immunofluorescent (IF) microscopy. This analysis revealed the numbers of nuclear YAP1 (nYAP1)-expressing HNF4A[+] hepatocytes, ACTA2[+] aHSC, and SOX9[+] ductular cells, were significantly reduced in *Scd2*[f/f]*;CC* (Fig. 1c, d). Further, HNF4A[+] HCC cells positive for nYAP1 were reduced in *Scd2*[f/f]*;CC* vs. *Scd2*[f/f] as quantified by 3-dimensional confocal microscopy imaging (Fig. 1e, f and Supplementary Fig. 1g). This global YAP1 repression in the multiple cell types including HCC cells was accompanied by repressed YAP1/TAZ-target genes such as *Ctgf*, *Cyr61*, and *Bric5* (Fig. 1g). *Cnnd1* and *Scd2* mRNA upregulation in *Scd2*[f/f], was also completely blunted in *Scd2*[f/f]*;CC*, likely reflecting global CTNNB1 inactivation in multiple

cell types caused by conditional *Scd2* knockout. IB analysis of liver tumors was difficult due to markedly diminished tumors in *Scd2*[f/f]*;CC*. However, qPCR on liver tumor RNA confirmed *Yap1* and *Wwtr1* repressions in *Scd2*[f/f]*;CC* (Supplementary Fig. 1h).

### CTNNB1 regulation of hepatic YAP1 and SCD expression
*Yap1* transcription is positively regulated by CTNNB1[12] and YAP1 and CTNNB1 were concomitantly repressed in *Scd2*[f/f]*;CC*, suggesting YAP1 repression may be caused by reduced CTNNB1. We also wondered if this regulation is unique to liver tumorigenesis or also relevant to normal liver. To this end, we analyzed YAP1 and *Scd1/2* expression in livers from *Ctnnb1*[f/f]*;AlbCre* mice lacking CTNNB1 primarily in hepatocytes but also in cholangiocytes. IB analysis revealed downregulation of YAP1, TAZ, HuR, and LRP6 in *Ctnnb1*[f/f]*;AlbCre* livers (Supplementary Fig. 1i). *Scd1/2*, the putative targets of CTNNB1[13] and *Yap1/Wwtr1* mRNA were also repressed (Supplementary Fig. 1j, k), suggesting that CTNNB1 positively regulates YAP1/TAZ and SCD-HuR-LRP6 pathway even in normal hepatocytes.

### aHSC SCD2 deficiency reduces LTB4R2 and its oxylipin ligands
SCD2 is an enzyme essential for generation of MUFAs which give rise to polyunsaturated fatty acids (PUFA) via elongation and desaturation. We, therefore, performed a lipidomic analysis for PUFA metabolites in *Scd2*[f/f]*;CC* vs. *Scd2*[f/f] livers. Among the major metabolites, *Scd2*[f/f]*;CC* had selective reductions in 12-hydroxyheptadecatrienoic acid (12-HHTrE), a 17-carbon PUFA of ill-defined sources; 12-hydroxyeicosatetraenoic acid (12-HETE) a 20-carbon PUFA derived from arachidonic acid; 9- and 13-hytroxyoctadecadienoic acid (9-HODE and 13-HODE) and 9,10-epoxyoctadecenoic acid (9,10-EpOME) derived from linoleic acid (Fig. 1h and Supplementary Data 3). We were intrigued by the reductions in 12-HHTrE and 12-HETE which are ligands for leukotriene B4 receptor 2 (LTB4R2) implicated in several malignancies[11–13] but not HCC to date. We also found LTB4R2 protein upregulation in *Scd2*[f/f] ATL was prevented in *Scd2*[f/f]*;CC* (Fig. 1i). Similarly, *Ltb4r2* mRNA expression in tumors was reduced in *Scd2*[f/f]*;CC* vs. *Scd2*[f/f] (Supplementary Fig 1h). Collectively, these results suggest that aHSC SCD2 controls the expression of LTB4R2 and its highest affinity oxylipin ligand, 12-HHTrE, and reduced LTB4R2 activation might have suppressed tumorigenesis in *Scd2*[f/f]*;CC* mice.

### LTB4R2 mediates YAP1 and CTNNB1 activation
To test the role of LTB4R2 in HCC, we examined the effects of pharmacologic and genetic inhibition of LTB4R2 on the growth of the human HCC cell line Huh7. The LTB4R2 antagonist LY255283, concentration-dependently inhibited Huh7 growth (Fig. 2a). Expression of shRNA against *LTB4R2* (*LTB4R2-shRNA*) reduced *LTB4R2* mRNA by 80% but not *LTB4R1* mRNA (Supplementary Fig. 2a). This knockdown (KD) reduced LTB4R2 protein and the cell growth (Supplementary Fig. 2b-c), confirming the growth-promoting role of LTB4R2. LTB4R2 KD decreased nYAP1 and CTNNB1 (Fig. 2b). Phosphorylation of YAP1 at S127 by LATS kinase results in YAP1 cytosolic retention and YAP1 is activated by LATS phospho-inhibition via alternative Wnt-FZD/ROR-Gα12/13-Rho GTPases pathway[14]. However, cytosolic p(S127)YAP1 was not different in LTB4R2 KD cells (Fig. 2b), suggesting the reduced nYAP1 was not due to LATS1/2-mediated post-translational regulation. *YAP1* mRNA was also repressed in LTB4R2 KD cells (Supplementary Fig. 2d), suggesting pre-translational downregulation as observed in *Scd2*[f/f]*;CC* mice (Supplementary Fig. 1h). Indeed, LTB4R2 KD decreased YAP1 in whole cell extracts (Supplementary Fig. 2e) which likely resulted in nYAP1 reduction. Inhibitory S9 phosphorylation of GSK3β was reduced in LTB4R2 KD cells (Fig. 2b), suggesting increased GSK3β substrate binding might have enhanced CTNNB1 proteasomal degradation. LTB4R2 KD abrogated phospho-ERK1/2 (p44/42 MAPK) (Fig. 2c), known to

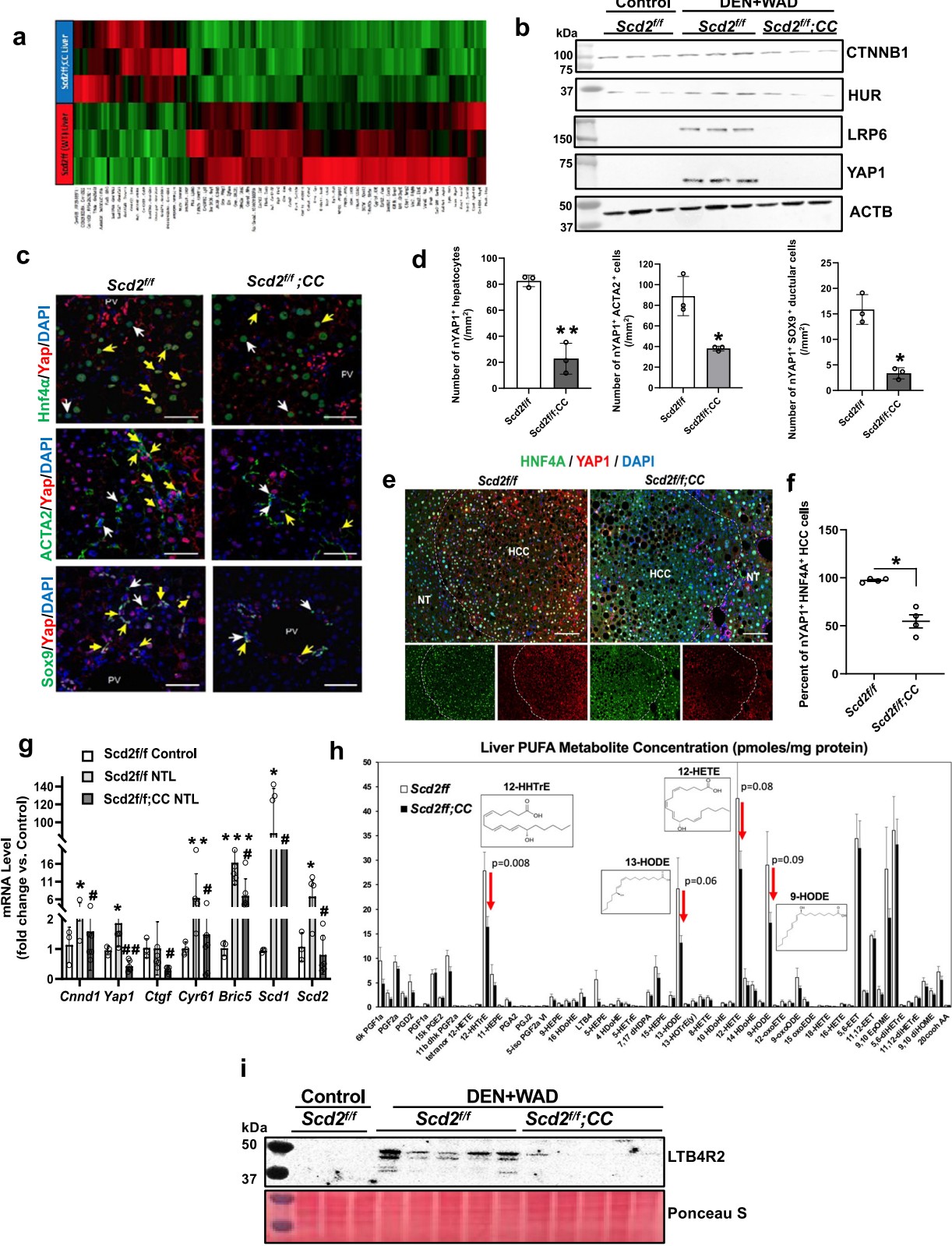

phosphorylate GSK3β at S9[15]. IF microscopy confirmed a significant reduction in nYAP1[+] by LTB4R2 KD (Fig. 2d). The LTB4R2 antagonist suppressed TEAD-luciferase reporter activity (Supplementary Fig. 2f) and the YAP1-target gene *CTGF* (Supplementary Fig. 2g). Similarly, LTB4R2 KD suppressed *CTGF* along with *CYR61* and *BRIC5* (Supplementary Fig. 2d). As expected from reduced nuclear CTNNB1, *CCND1* mRNA was decreased by LTB4R2 KD or

LY255283 treatment (Supplementary Fig. 2d and g). *LTB4R2* mRNA was also reduced by LY255283, suggesting a positive forward regulation of this receptor.

## 12-HHTrE upregulates YAP1 via LTB4R2 and CTNNB1
Conversely, the LTB4R2 ligand 12-HHTrE stimulated TEAD promoter activity (Fig. 2e), an effect which was lost by LTB4R2 KD or co-

**Fig. 1 | SCD2 deficiency in aHSC results in global repression in CTNNB1, YAP1, LTB4R2, and its ligands. a** A heatmap of DEGs in tumor-adjacent liver (TAL) of *Scd2[f/f];Col1a1Cre (CC)* vs. *Scd2[f/f]* mice subjected to the DEN + WAD regimen (*n* = 3 mice per group). **b** IB analysis of TAL proteins from *Scd2[f/f];CC* vs. *Scd2[f/f]* mice as compared to *Scd2[f/f]* control mice. (*n* = 3 mice per group). **c** IF microscopy of nYAP1⁺HNF4A⁺ hepatocytes, nYAP1⁺ACTA2⁺ aHSC, nYAP1⁺SOX9⁺ ductular cells in *Scd2[f/f];CC* vs. *Scd2[f/f]* mouse TAL. Scale bar = 50 µm. White arrows point the cells with HNF4A⁺, ACTA2⁺, or SOX9⁺ staining while yellow arrows point the cells with the dual staining with nYAP1. **d** Morphometric analysis of nYAP1⁺ hepatocytes, aHSC, and ductular cells. **p* < 0.05 and ***p* < 0.01 vs. *Scd2[f/f]* mouse TAL by two-sided t-test. Data presented as means ± SEM (*n* = 3 different sections). Exact p values are shown in the Source Data File. **e** Co-IF microscopy of nYAP1⁺HNF4α⁺ liver tumor cells (HCC). Scale bar = 100 µm. The border between HCC and non-tumorous (NT) areas is indicated by a broken line. Images shown are representative of four pairs of samples analyzed. **f** Imaging morphometric data for the percentage of nYAP1⁺HNF4α⁺ liver tumor cells by 3-dimensional confocal microscopy analysis. **p* < 0.05 vs. *Scd2[f/f]* by two-sided t-test. Data shown are means ± SEM (*n* = 4 pairs of samples). **g** qPCR data for *Scd2[f/f];CC* vs. *Scd2[f/f]* mouse TAL (*n* = 6 each) compared to *Scd2[f/f]* control normal liver (*n* = 3). **p* < 0.05, ***p* < 0.01, ****p* < 0.005 vs. *Scd2[f/f]* control; #*p* < 0.05 and ##*p* < 0.01 vs. *Scd2[f/f]* TAL by two-sided t-test. **h** Lipidomic analysis for PUFA metabolites in *Scd2[f/f];CC* vs. *Scd2[f/f]* mouse TAL. Data presented as means ± SEM (*n* = 3 mouse samples per group). *P* values determined by two-sided t-test. Red arrows depicting reductions in four specific metabolites in *Scd2[f/f];CC* TAL (Raw data are provided in Supplementary Data File). **i** IB analysis of LTB4R2 for *Scd2[f/f];CC* (*n* = 6 mice) vs. *Scd2[f/f]* (*n* = 5 mice) TAL proteins compared to *Scd2[f/f]* control mice (*n* = 3 mice). For all relevant figures, source data and exact *p* values are provided in the Source Data file.

treatment with the Super-TDU, an inhibitor of YAP-TEAD interaction (Supplementary Fig. 2h, i). 12-HHTrE increased nYAP1⁺ cells by IF (Fig. 2f) and nYAP1 protein by IB with no changes in cytosolic p(S127) YAP1 (Fig. 2g). 12-HHTrE increased cytosolic pERK1/2 and p(S9)GSK3β, nuclear CTNNB1 (Fig. 2g) and upregulated mRNAs for *YAP1*, YAP1-target genes (*CTGF*, *CYR61*), CTNNB1-target genes (*CCND1*, *LGR5*), and the antiapoptotic gene *BCL2L1* activated by YAP1-TBX5-CTNNB1 complex[16] (Supplementary Fig. 2j). 12-HHTrE upregulated LTB4R2 mRNA and protein (Supplementary Fig. 2j-k), supporting the positive feed-forward regulation. These results are opposite of the LTB4R2 inhibition effects and enforce the notion that 12-HHTrE-LTB4R2 signaling drives YAP1 and CTNNB1 activation. CTNNB1 upregulates *YAP1* transcription by its binding to TCF4 recruited to the *YAP1* intronic enhancer[17]. Indeed, 12-HHTrE activated *YAP1* enhancer-luciferase reporter in Huh7 cells, and a mutation of the TCF element abrogated this activation (Fig. 2h). ChIP-qPCR analysis showed 12-HHTrE increased co-enrichments of CTNNB1 and RNA polymerase II (RNA-PII) at the region encompassing this TCF site (YAP-Y3, Fig. 2i) and the proximal promoter region (YAP-Y2). Further, CTNNB1 KD abrogated 12-HHTrE-induced *YAP1* mRNA (Supplementary Fig. 2l). CTNNB1 KD also prevented 12-HHTrE-mediated *LTB4R2* induction (Supplementary Fig. 2m), suggesting the role of CTNNB1 in the positive forward regulation of the receptor. Using *LTB4R2* proximal promoter-first intron (−1517/+344) deletion constructs with a luciferase reporter we have cloned, a −492/−244 region and a +258/+344 intronic region, were shown responsible for the 12-HHTrE stimulation (Fig. 2j) and these activities were attenuated by CTNNB1 KD (Supplementary Fig. 2n).

## aHSC produces 12-HHTrE to activate LTB4R2-CTNNB1-YAP1 signaling in HCC

We next asked how SCD2 in aHSC supported YAP1 activation in HCC cells via the LTB4R2 pathway. For this, we treated Huh7 cells with conditioned medium (CM) from the human aHSC LX2 cells transduced with SCD-shRNA (SCD-sh) vs. scrambled shRNA (SCR-sh). Here it should be noted that only SCD1 isoform or simply referred to SCD, is mainly expressed in LX2 cells. LX2 SCR-sh CM but not SCD-sh CM, increased TEAD promoter activity in Huh7 cells (Fig. 3a). The LX2 CM-induced TEAD activation required LTB4R2 and YAP1 as the receptor antagonist LY255283 and Super-TDU abrogated the effect (Supplementary Fig. 3a, b). Using a size-exclusion column, the TEAD promoter stimulatory activity was shown to exist in a <10 kDa filtrate (Supplementary Fig. 2c). Lipid removal from the <10 kDa filtrate with the lipid adsorption reagent (Cleanascite™) abolished the activity (Supplementary Fig. 3d), suggesting lipid LTB4R2 ligands are released SCD-dependently by aHSC to activate LRB4R2-YAP1 pathway in HCC cells. However, we also showed that LTB4R2 antagonism reduced basal TEAD activity in Huh7 cells (Supplementary Fig. 2f), suggesting spontaneous release of LTB4R2 ligands, which may contribute to the LX2 CM effect. To test this possibility, we treated Huh7 cells with LX2 CM in the presence of the thromboxane A synthase 1 (TBXAS1) inhibitor

Ozagrel vs. vehicle (DMSO). TBXAS1 was suspected as a major 12-HHTrE biosynthetic enzyme in Huh7 cells based on its high mRNA expression (not shown). Indeed, Ozagrel suppressed the basal TEAD activity by 50%. Yet, the LX2 CM still increased the activity in both DMSO and Ozagrel-treated cells to a similar extent (Fig. 3b), suggesting the autocrine LTB4R2-TEAD activation in Hur7 cells is not required for paracrine stimulation by LX2-CM. However, LX2 CM upregulated *PTGS2* and *TBXAS1* in Huh7 cells and this effect was lost by SCD KD in LX2 cells (Supplementary Fig. 3e), suggesting aHSC paracrine effect may support autocrine release of the LTB4R2 ligand 12-HHTrE.

## CYP1B1 contributing to aHSC generation of 12-HHTrE. CYP450 enzymes such as TBXAS1 (CYP5A1), CYP1A1, and CYP1B1 generates 12-HHTrE from prostaglandin G2 and H2[17,18]. Although platelets are considered a major source of 12-HHTrE[19], it is also generated by macrophages[20], keratinocytes[21], and malignant cells[11–13]. As conditional *Scd2* ablation in *Scd2[f/f];CC* mice reduced 12-HHTrE concentration in TAL, we screened RNA-seq DEGs of *Scd2[f/f];CC* vs. *Scd2[f/f]* TAL for 12-HHTrE-biosynthetic CYP genes. This revealed that *Cyp1b1*, but not other *Cyp* genes, was significantly (*p* = 5.15E-3) repressed 4.5-fold in *Scd2[f/f];CC*, suggesting the role of CYP1B1 in regulating the liver oxylipin level. We next examined which cell types express *Cyp1b1* in TME by screening the cell type clusters identified by scRNA-seq as described above (Supplementary Fig. 1a). As shown by violin plots (Fig. 3c), *Cyp1b1* was dominantly expressed by *Fbln2⁺* cells. To further characterize these *Fbln2⁺ cells*, we enriched *Col1a1*-expressing cells by isolating GFP⁺ cells by FACS from DEN + WAD-subjected *Rosa26mTmG* reporter mice (*Gt(ROSA)26Sor[tm4(ACTB-tdTomato,-EGFP)Luo]*) carrying *Col1a1-Cre* (*mTmG;CC*) based on the gating strategy depicted in Supplementary Fig. 3f. This allowed separation of VitA⁺GFP⁻ quiescent HSC (blue box), VitA⁺GFP⁺ aHSC (red box), and VitA⁻GFP⁺ VitA-depleted aHSC and activated PF (green box) (Fig. 3d). In normal mouse, VitA⁺GFP⁺ quiescent HSC was a major population while in DEN + WAD mouse, VitA⁺GFP⁺ aHSC and VitA⁻GFP⁺ cells expanded reflecting activation of HSC (Fig. 3d and Supplementary Fig. 3g). By subjecting the latter two fractions to scRNA-seq, we revealed that *Cyp1b1* but not other biosynthetic genes, was expressed by a subset of VitA⁺GFP⁺ aHSC which emerged in the DEN + WAD liver (Fig. 3e, top). This subset co-expressed the HSC marker *Lrat* and the PF marker *Fbln2* but not another PF marker *Thy1*, labeled as *Lrat⁺Thy1⁻Fbln2⁺* (L+T-F+, Fig. 3f). In VitA⁻GFP⁺ cells, only *Cyp1b1* was expressed in both *Lrat⁺Fbln2⁺* and *Lrat⁻Fbln2⁺* cells in small numbers in control liver but *Cyp1b1⁺* cells increased in the DEN + WAD liver (lower panel of Fig. 3e and Fig. 3g) with *Lrat⁺Cyp1b1⁺* cells being 3x more abundant than *Lrat⁻Cyp1b1⁺* cells (Supplementary Fig. 3i). Collectively, these results demonstrated that VitA⁺ or VitA⁻ *Lrat⁺Fbln2⁺* aHSC was a unique tumor-associated aHSC subpopulation which likely produced 12-HHTrE by CYP1B1. The potential importance of the L+T-F+ aHSC was also supported by their selective expression of tumor promoter genes such as *Ereg*[22], *Vgefa*[22], *Gas6*[23], and *Mmp2*[24] (Supplementary Fig. 3h). To scrutinize the

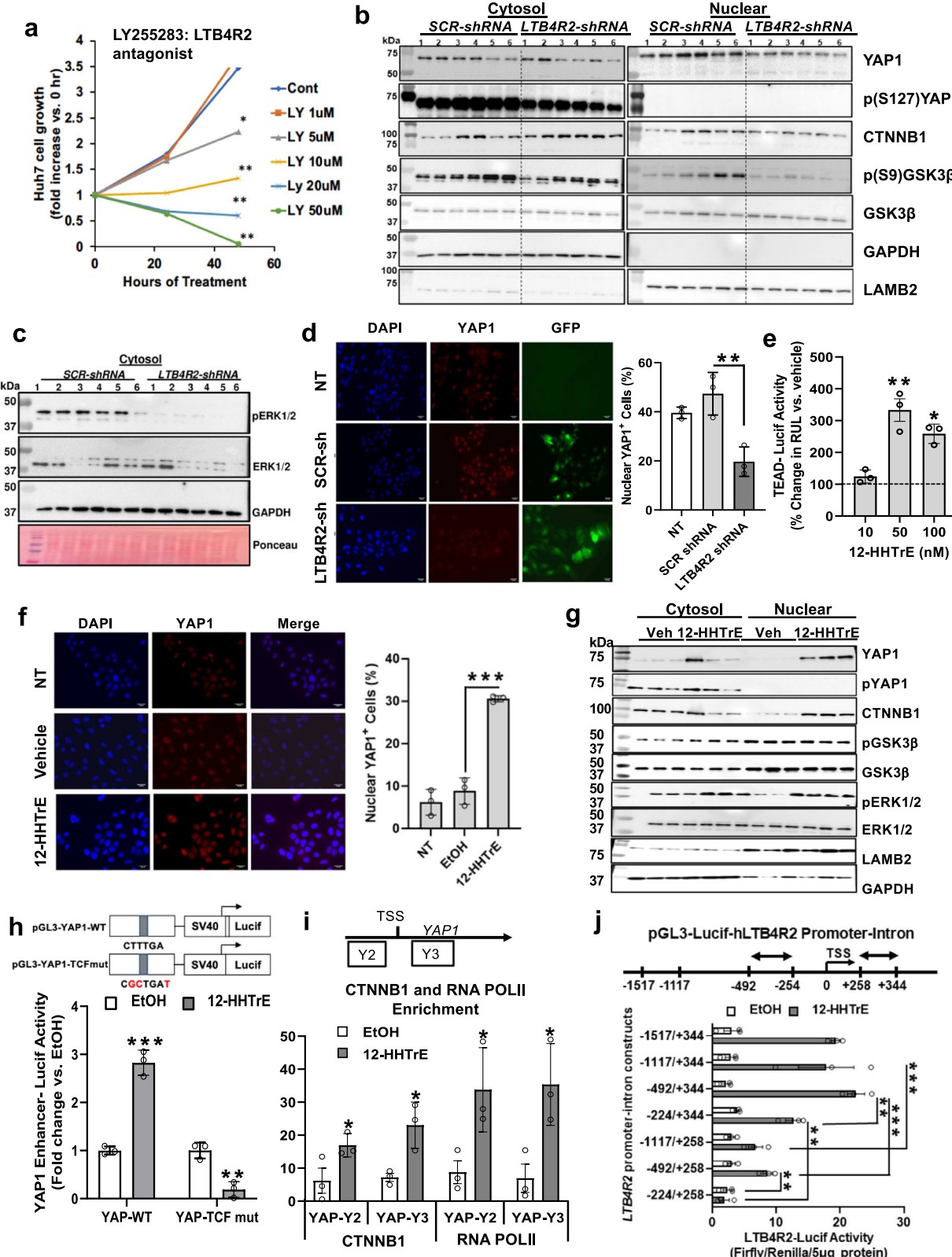

functional significance of CYP1B1 expressed by aHSC, we silenced *CYP1B1* by CRISPR/Cas9 in LX2 cells by using two different guide RNAs (sgRNA-A and -B) (Fig. 3h). This manipulation reduced the release of oxylipins into the media including 12-HHTrE (Fig. 3i and Supplementary Data 3) and blunted the CM-mediated upregulation of the TEAD promoter activity (Fig. 3j).

To extend our results to human HCC, we analyzed human HCC scRNA-seq data available from NCI's Single-cell Atlas in Liver Cancer (scATlasLC). Among different cell types in the human HCC TME, *CYP1B1* was expressed by CAF, TAM (tumor-associated macrophages), and TEC (tumor-associated endothelial cells). TBXAS1 expression was prominent in TAM and T cells but not in CAF (Supplementary Fig. 3j). These results suggested that TAM may be a major source of the oxylipin in human

**Fig. 2 | 12-HHTrE-LTB4R2 signaling activates CTNNB1 and YAP1 and auto-induces LTB4R2. a** Concentration-dependent suppression of Huh7 cell growth by the LTB4R2 antagonist LY255283. *$p < 0.05$, **$p < 0.01$ vs. vehicle control by two-sided t-test ($n = 3$ separate experiments). **b** IB analysis of cytosolic and nuclear proteins from Huh7 cells with LTB4R2 KD (*LTB4R2-shRNA*) vs. control (*SCR-shRNA*). The numbers shown above the lanes indicate repeated KD experiments ($n = 6$). **c** IB analysis for pERK1/2 and ERK1/2 of cytosolic proteins from *LTB4R2-shRNA* vs. *SCR-shRNA* Huh7 cells as described above ($n = 6$ experiments). **d** IF microscopy for nYAP1 of Huh7 cells cultured in ~40% density and infected with adenovirus expressing EGFP plus *LTB4R2-shRNA* or *SCR-shRNA* with morphometric data for the percentage of nuclear YAP1+ cells. NT is a no infection control. Data presented as means ± SEM ($n = 3$ separate experiments). **$p < 0.01$ vs. SCR-shRNA cells by two-sided t-test. **e** TEAD-luciferase activity as the readout of YAP activity in Huh7 cells treated with 12-HHTrE. Data presented as means ± SEM ($n = 3$ separate experiments). *$p < 0.05$, **$p < 0.001$ vs. vehicle treatment (dotted line) by one-way post-hoc ANOVA test. **f** IF microscopy of nuclear YAP1 in Huh7 cells cultured in higher (~65%) density in serum-free media without (NT) or with the treatment of vehicle or 12-HHTrE (50 nM). A bar graph shows morphometric data for the percentage of nYAP1+ cells ($n = 3$ separate experiments). Data presented as means ± SEM. ***$p < 0.001$ vs. vehicle by two-sided t-test. **g** IB analysis of cytosolic and nuclear proteins from Huh7 cells treated with vehicle (Veh.) or 12-HHTrE (50 nM) ($n = 3$ separate experiments). **h** 12-HHTrE activation of wild-type vs. TCF site-mutated YAP1 intronic enhancer as measured by luciferase reporter activity. Mutated nucleotides in the enhancer are shown in red. **$p < 0.01$ and ***$p < 0.001$ vs. vehicle EtOH by two-sided t-test. Data presented as means ± SEM ($n = 3$ separate experiments). **i** ChIP-qPCR data for enrichment of CTNNB1 and RNA polymerase II (RPOLII) at the proximal promoter region (Y2) and the first intronic enhancer region (Y3). *$p < 0.05$ vs. EtOH by two-sided $t$ test. Data presented as means ± SEM ($n = 3$ separate experiments). **j** 12-HHTrE-stimulated *LTB4R2* proximal promoter-first intron activities with various deletions as determined by a luciferase reporter in Huh7 cells. **$p < 0.01$ and ***$p < 0.001$ vs. vehicle by two-sided t-test. Data presented as means ± SEM ($n = 3$ separate experiments). For all relevant figures, source data and exact $p$ values are provided in the Source Data file.

HCC TME and CYP1B1 as the oxylipin source in CAF as in our mouse model. Further, co-expression analysis showed, CYP1B1-expressing CAF are mostly FBLN2-positive (Supplementary Fig. 3k).

### LTB4R2 KD reproduces tumor suppression of conditional SCD2 deficiency

We next tested the causal role of LTB4R2 in liver tumorigenesis in vivo by administration of an AAV8 vector expressing shRNA against *Ltb4r2* vs. scrambled shRNA in DEN-WAD B6 mice at 1 month prior to sacrifice. Both AAV8 viral vectors equally transduced hepatocytes at ~60% as shown by the reporter GFP expression (Supplementary Fig. 4a) including AFP+ tumor cells (Supplementary Fig. 4b). This KD manipulation decreased hepatic LTB4R2 protein and nuclear CTNNB1 and YAP1 (Fig. 4a) and *Ltb4r2* mRNA in liver tumors (Supplementary Fig. 4c). The KD appeared to reduce liver weight/body weight ratio (Supplementary Fig. 4d) and significantly suppressed tumor multiplicity and total tumor volume (Fig. 4b), reproducing the phenotype observed in the *Scd2f/f;CC* mice and validating the role of LTB4R2 in CTNNB1 and YAP1 activation and liver tumorigenesis in vivo.

### Translational relevance

To attain the translational relevance of our findings, we analyzed the TCGA-LIHC cohort data to examine the relationship between *LTB4R2* expression per RNA-seq and patient survival. As our results suggested the role of CTNNB1 in YAP1 and LTB4R2 regulation and the gain-of-function *CTNNB1* mutation is common in HCC, we stratified the patients by the CTNNB1 missense mutation status. In either patient group with (92 patients) or without (263 patients) the mutation, no significant difference in survival between patients with high vs. low *LTB4R2* expression was observed (Supplementary Fig. 4e). However, our qPCR analysis revealed that the expression of 12-HHTrE biosynthetic genes (*PTGS2, TBXAS1, CYP1B1*), and *YAP1* and *LTB4R2* were all significantly upregulated in patient HCC vs. normal liver (Fig. 4c). We also performed IHC for CYP1B1 and LTB4R2 in patient HCC sections and detected CYP1B1+ CAF surrounding LTB4R2+ HCC cells (Fig. 4d), a spatial relationship that supports aHSC-HCC crosstalk involving the CYP1B1-LTB4R2 pathway. Finally, we used patient HCC organoids to test the effects of the LTB4R2 antagonist treatment and LTB4R2 KD. For this study, we used two different HCC organoids: model-1 expressing 30-fold higher *LTB4R2* mRNA compared to model-2. The model-1 organoid grew 160% faster than model-2 and showed more conspicuous growth inhibition by LTB4R2 antagonism or KD as compared to model 2 (Fig. 4e), suggesting the therapeutic potential of targeting the 12-HHTrE-LTB4R2 pathway for growth inhibition of HCC.

## Discussion

LTB4R2 was discovered as a low-affinity receptor for LTB4 in 2000[25–28], followed by the identification of 12-HHTrE as a high-affinity LTB4R2 ligand[29]. LTB4R2 is implicated in malignancies via its activities toward activation of ERK1/2, STAT3, and NF-κB[30–34]. Our study describes the role of LTB4R2 in liver tumor development via activation of CTNNB1 and YAP1, the two transcriptional activators that promote liver tumorigenesis[32,33]. We suggest that LTB4R2 activation by 12-HHTrE produced by aHSC CYP1B1 causes CTNNB1 activation via GSK3β pathway and subsequent *YAP1* transcription and tumor promotion (Fig. 4g). Our scRNA-seq analysis identified a *Lrat+Fbln2+* aHSC sub-population as the putative source of CYP1B1-catalyzed 12-HHTrE generation, the finding that was supported by the presence of CYP1B1+ CAF in patient HCC via IHC and scRNA-seq analysis. The tumor-promoting role of aHSC SCD-CYP1B1-tumor LTB4R2 pathway was supported by tumor suppression in mice by conditional *Scd2* ablation and AAV8-based *Ltb4r2* KD. Mechanisms for SCD2-dependent *Cyp1b1* upregulation are currently unknown but may involve its transcriptional activation by CTNNB1-AhR as shown for *Cyp1a1*[34].

Our TCGA-LIHC cohort analysis failed to show a positive association between LTB4R2 expression and poor survival. We do not know the reasons for this seemingly discrepant result. The LTB4R2 pathway we disclosed may be active only in focal areas of growing HCC and may not be accurately assessed by RNA-seq. Indeed, our RNA-seq analysis in *Scd2f/f;CC* mice failed to detect reduced *Ltb4r2* expression (Supplementary. Table 1) despite significant reductions shown by IB or qPCR (Fig. 1i and Supplementary Fig. 1h). A population of CYP1B1-expressing aHSC in TME is relatively small (Fig. 3e and Supplementary Fig. 3j, k) but likely upregulates LTB4R2 in adjacent tumor cells via paracrine positive-forward regulation of the receptor that we disclosed. Thus, only those HCC cells surrounded by CYP1B1+ aHSC, may express LTB4R2 (Fig. 4d) - the unique focal and spatial nature of the CYPB1-LTB4R2 pathway. Despite this negative result, our three different translational approaches including patient HCC qPCR and IHC analyses as well as the patient HCC organoid study, consistently supported the relevance and functionality of the LTB4R2 pathway.

Remarkably, conditional SCD deficiency selectively achieved in aHSCs, repressed YAP1 in multiple cell types, suggesting the 12-HHTrE-LTB4R2 pathway has a broader impact on TME. In this regard, the conditional *Scd2* knockout might have indirectly impacted oxylipin generation by other cell types such as TAM or TEC shown to express *CYP1B1* or/and *TBXAS1* per the human HCC scRNA-seq data. As these cells promote tumor cell growth, migration, and metastasis[35,36], 12-HHTrE released by aHSC may serve as an upstream event. Lipid metabolic reprogramming is an essential metabolic adaptation for cancer cell survival and growth[37] and activation of HSCs[38,39] characterized by the loss of vitamin A and lipid storage[40]. Tumor-promoting PUFA metabolism may include the ability of PGE2 released by tumor cells to render immunosuppressive effects via myeloid suppressor cell activation[41] and to stimulate neovascularization[42]. The 12-HHTrE-LTB4R2 pathway revealed by our study is a mode of SCD-dependent

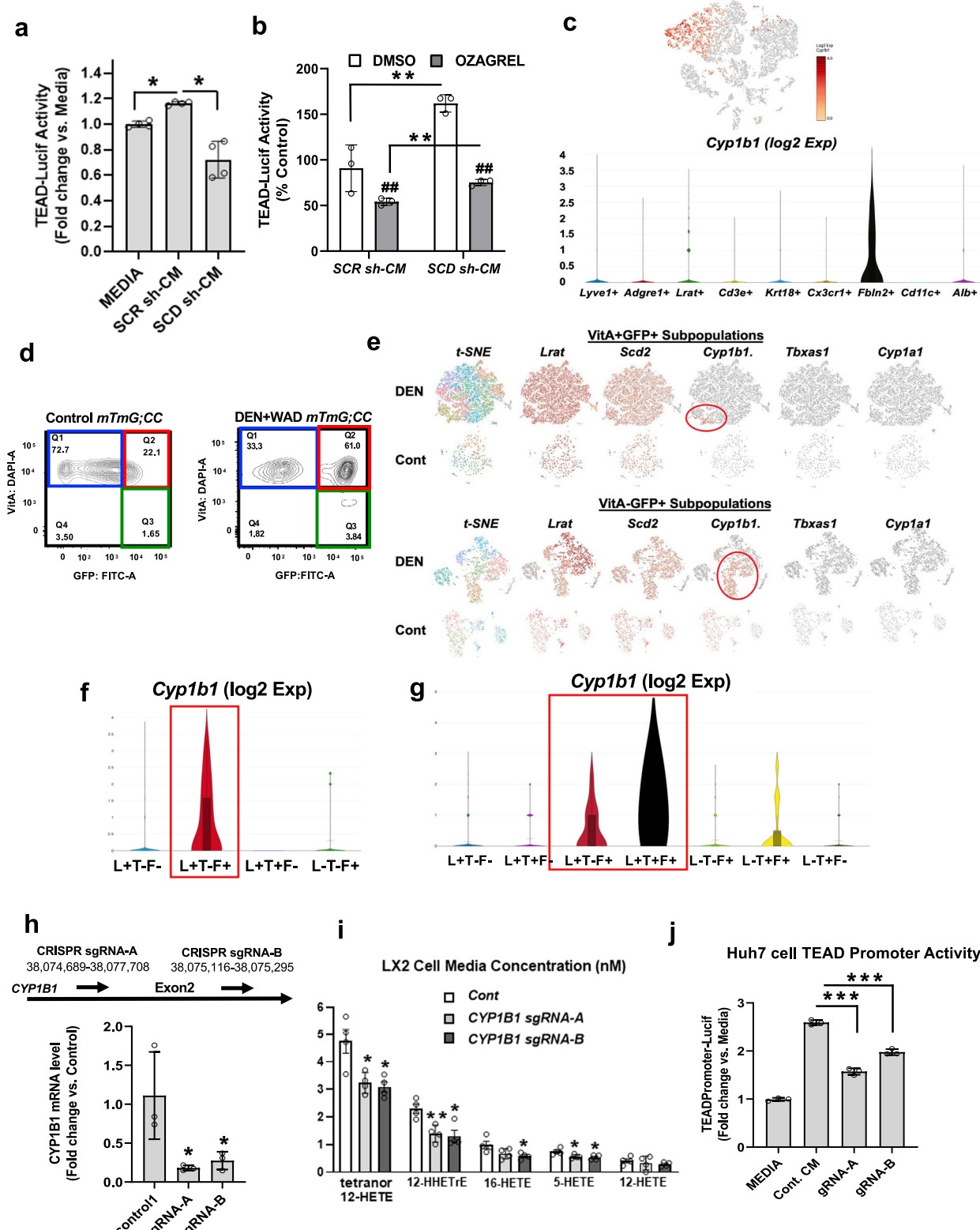

PUFA reprogramming in aHSC which has tumor-promoting paracrine effects likely involving similar reprogramming in tumor cells and other TME cell types. A recent study utilizing scRNA-seq and genetic manipulations, revealed HSC subpopulations with HCC protective vs. promoting function and a shift in the subpopulations associated with increased HCC risk[43]. It also highlighted the importance of HSC-mediated tumor promotion occurring in non-tumor, tumor-adjacent liver tissues as our research described. The *Cyp1b1*[+] subpopulation we identified may participate in this HCC-promoting shift.

## Methods

### Ethical approval

This study was conducted according to the National Institute of Health guidelines and all animal experimentations and procedures were

**Fig. 3 | aHSC release LTB4R2 ligands in a manner dependent on CYP1B1. a** TEAD-luciferase activity in Huh7 cells treated with conditioned medium (CM) from LX2 cells transduced with *scrambled shRNA (SCRsh-CM)* vs. *SCD-shRNA (SCDsh-CM)* as compared to the media without Huh7 cells (Media). *$p < 0.05$ by two-sided t-test. Data presented as means ± SEM ($n = 4$ separate experiments). **b** TEAD-luciferase activity in Huh7 cells exposed to *SCRsh-CM* vs. *SCDsh-CM* in the presence of the TBXAS1 inhibitor Ozagarel or vehicle DMSO. *$p < 0.001$ vs. Media, #$p < 0.001$ vs. DMSO by two-sided t-test. Data presented as means ± SEM ($n = 3$ experiments). **c** A scRNA-seq t-SNE plot showing *Cyp1b1*[+] cells and violin plots revealing selective *Cyp1b1* expression by *Fbln2*[+] cells. Cell numbers for different cell type groups are provided in Supplementary Data 3. **d** Contour FACS plots of liver mesenchymal cells isolated from control vs. DEN + WAD treated *Rosa26mTmG;Col1a1-Cre; (mTmG;CC)* mouse, gated by DAPI (*Y*-axis) for Vit A fluorescence and FITC (*X*-axis) for Col1a1-GFP, revealing VitA[+]GFP[-] quiescent HSC (blue), VitA[+]GFP[+] aHSC (red), and VitA[-]GFP[-] cells (green). FACS gating strategies are provided in Supplementary Information. **e** scRNA-seq analysis for expression of 12-HHTrE biosynthetic genes in VitA[+]GFP[+] (top) and VitA[-]GFP[+] (bottom) subpopulations from DEN+WAD mouse (DEN) vs. normal (Cont.) livers. **f** Violin plots for *Cyp1b1* expression by subpopulations based on *Lrat*, *Thy1*, and *Fbln2* expression in VitA[+]GFP[+] cells from the DEN mouse and **g** in VitA[-]GFP[+] cells. (See Supplementary Data 3 for parameter values for violin plots and cell numbers for different subpopulations). **h** CRISPR/Cas9 ablation of CYP1B1 in LX2 cells using the guide RNA-A (sgRNA-A) or -B (sgRNA-B) (top), represses CYP1B1 mRNA. *$p < 0.05$ vs. control by two-sided t-test. Data presented as means ± SEM ($n = 3$ separate samples). **i** Oxylipin concentrations in CM from LX2 cells with CYP1B1 KD are described above. *$p < 0.05$ vs. control by two-sided t-test. Data presented as means ± SEM ($n = 4$ separate samples). (Raw data provided in Supplementary Data 4 in Supplementary File). **j** Reduced stimulatory effects of CM from CYP1B1 ablated LX2 cells on the TEAD promoter activity in Huh7 cells (right). ***$p < 0.001$ vs. Control CM by two-sided t-test. Data presented as means ± SEM ($n = 3$ experiments). For all relevant figures, source data and exact $p$ values are provided in the Source Data file.

approved by the Institutional Animal Care and Use Committee of the University of Southern California (# 20426). Male mice were used for the present study as the male sex is predisposed to HCC development[44]. Patient or normal liver tissues were obtained from existing biorepositories under the approved IRB protocols of the University of Southern California (HS-16-00392), Kansas University Medical Center (11378), Johns Hopkins University (00107893), McGill University (11-066 SDR), and Mt Saini School of Medicine (20-04150). Informed consent was obtained under each protocol for original deposition of specimens and their distribution for biomedical research. No additional consent was required for our use of qPCR, immunohistochemical, and organoid analyses. All animals are housed as per approved housing conditions by Institutional Animal Care and Use Committee of the University of Southern California, including humidity range from 30 to 70%, temperature range from 20 °C to 26 °C and day/light cycle on time 6 a.m. off time 6 pm.

## Animal experiments

The conditional knockout male mice carrying ablated *Scd2* in *Col1a1*-expressing activated HSCs (aHSC) on C57BL/6J background were generated by paired breeding over at least 7–8 generations of mice in which third exon of *Scd2* was flanked by *loxP* site (*Scd2[f/f]*) with mice expressing the *Cre* recombinase transgene under the promoter of *Col1a1* (CC) as previously described[6]. The resultant male *Scd2[f/f];CC* mice and wild type *Scd2[f/f]* mice were injected with DEN (10 mg/kg) at the age of 2 weeks and fed for 5 months starting 6 weeks old with liquid Western alcohol diet (WAD: Dyets, Inc. #710362) containing lard (23.2 g/L), cholesterol (2.32 g/L), and ethanol (3.5%v/v). Tumor and TAL tissues were collected for lipidomic, RNA-sequencing, quantitative polymerase chain reaction (qPCR), immunofluorescence microscopy (IF), and immunoblotting (IB) analyses. Adeno-associated virus serotype 8 (AAV8) expressing shRNA against *LTB4R2* or scrambled shRNA were injected at $5 \times 10^{11}$ GC per mouse via tail vain of C57/B6 male mice under DEN-WAD regimen one month prior to sacrifice. Tumor number and dimension were measured using a caliper and tumor volume was estimated using the formula of tumor volume (mm³) = $(D \times d^2)$ ½, where *D* is longest diameter and d is the shortest diameter. TAL and tumor tissues were removed, snap-frozen and stored at −80 °C for further RNA and protein analysis. As per approved tumor size/burden permitted by Institutional Animal Care and Use Committee of the University of Southern California, if the combined volume of all tumors exceeds 2000 mm³ or any single tumor meets the above criteria, mouse will be considered as end point of study. If a mouse also appears in distress, regardless of the size of the tumor or the weight of the animal and its ability to eat and drink is affected or impaired and weight loss exceeding 20% of the body weight then consider that as endpoint of the study.

## Cell isolation and culture experiments

For isolation of *Col1a1*-expressing liver mesenchymal cells, *Rosa26mTmG* reporter mice (*Gt (ROSA)26Sortm4(ACTB-tdTomato, EGFP) Luo*) carrying *Col1a1-Cre* (*mTmG;CC*) on C57BL/6J background were generated by successive paired breeding and males were subjected to the DEN-WAD regimen, livers were perfused with pronase and collagenase and GFP+ *Col1a1*-expressing cells were collected via FACS[6,29]. The cells were sorted on ARIA IIu (Becton Dickinson) with solid state laser excited at 488 nm and measured with 510/20BP emission mirrors for GFP and excited at 355 nm and measured with 450/50BP mirrors for vitamin A and data were analyzed by FlowJo 10.8.1 version. The vitamin A gating was set with freshly mouse primary HSCs as the positive control and the rat myofibroblast cell line BSC as the negative control. Col1a1-GFP gating was set with freshly isolated normal C57BL/6 mouse HSCs as the negative control and fully culture-activated HSCs from Col1a1-GFP mice as the positive control. Examples of these gating strategies are described in Supplementary Information. Human HCC cell line Huh7 cells (cat# 01042712, Sigma-Aldrich) were treated with the LTB4R2 antagonist LY255283 (10 μM, cat#70715, Cayman Chemical Company), 12-HHTrE (50-100 nM, cat#34590, Cayman Chemical Company) or vehicle (ethanol or DMSO) in serum-free DMEM media for 24-48 hr for cell growth, mRNA and protein expression, and TEAD promoter-reporter analyses. The cells were also infected with adenovirus expressing *LTB4R2 shRNA* (PFU/ml = $8.6 \times 10^{10}$) vs. scrambled shRNA (PFU/ml= $1.1 \times 10^{11}$) (Vector Biolabs, Malvern, PA) at MOI = 15 in DMED growth media containing 10% FBS. The cells were then washed with 1X PBS after overnight, incubated in serum-free media for 48 hr, washed, harvested, and centrifuged at $4000 \times g$ for 10 min at 4 °C to snap freeze the cell pellets for subsequent protein and mRNA analysis. To study crosstalk of aHSC-HCC, Huh7 cells (cat# 01042712, Millipore Sigma) were treated with CM collected from the human aHSC LX2 cells (cat# SCC064, Fisher Scientific) infected with an adenovirus expressing SCD-shRNA vs. SCR-sh-RNA (Vector Biolabs) or subjected to lentiviral vector-based CRISPR/Cas9 expression to ablate *CYP1B1* by testing two gRNA sequences (GeneScript, cat number# SC1805, 1678).

## Lipidomic analysis

Targeted lipidomic analysis was performed by Lipidomic Core of University of California at San Diego, on snap-frozen mouse liver samples and conditioned media collected from LX2 cells with SCD-shRNA vs. SCR-sh-RNA expression and conditioned media from mouse primary HSCs treated with the SCD inhibitor vs. vehicle. Briefly, tissue samples were homogenized, sonicated for 6 s, and resuspended in 1.0 ml of 10% methanol:water (v/v). Samples were spiked with 50 μL of a 50 pg/μL (2.5 ng total) deuterated internal standard solution. Lipid metabolites were extracted using strata-x 33 u polymerized solid reverse phase extraction columns. Briefly, columns were washed with 3.5 ml of 100%

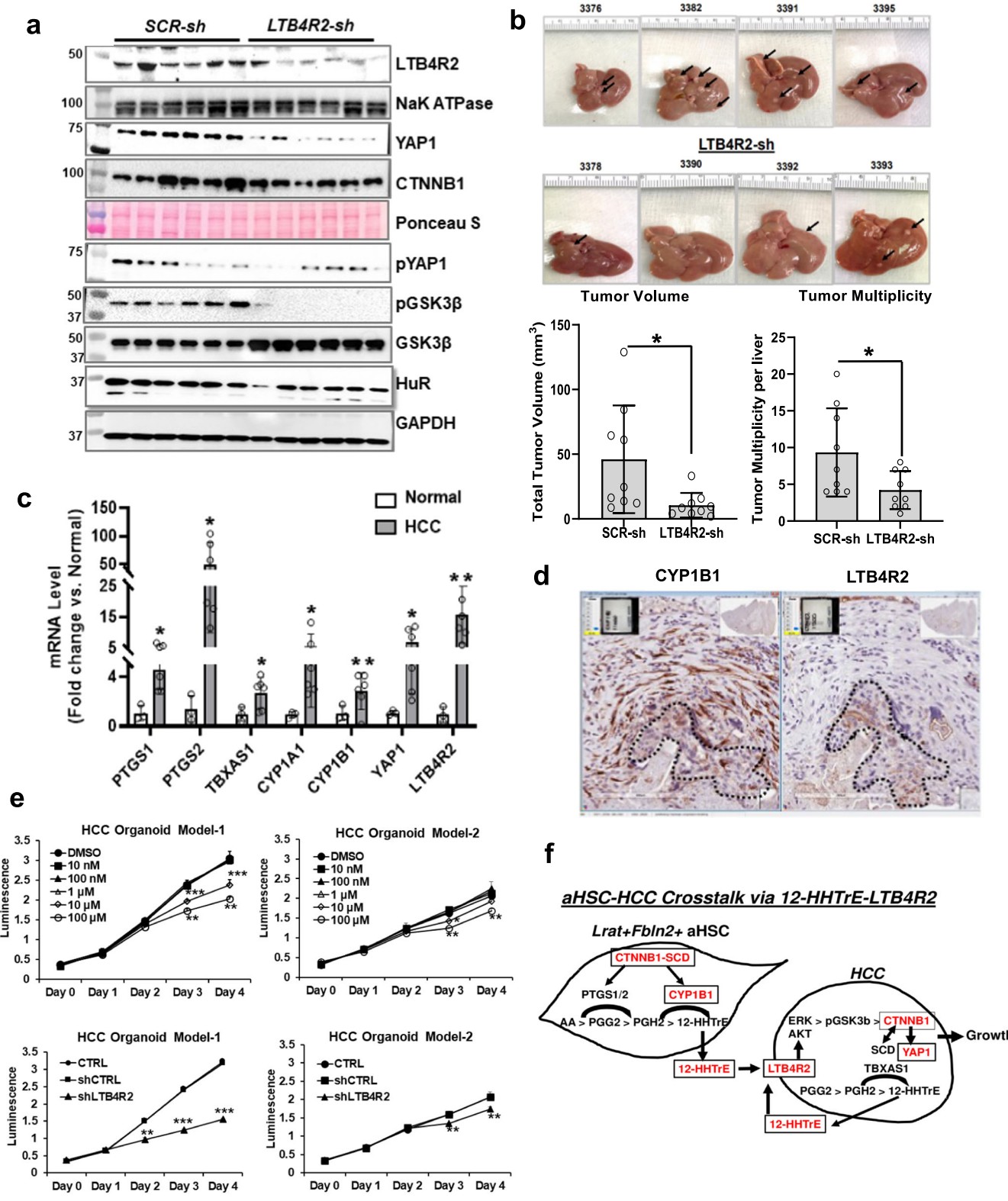

methanol, followed by 3.5 ml of water. Samples were then washed with 3.5 ml of 10% methanol to remove any non-specific binding metabolites. The eicosanoids were then eluted with 1 ml of 100% methanol, and the eluent was dried under vacuum, dissolved in 50 µl of buffer A consisting of water-acetonitrile-acetic acid, 60:40:0.02 (v/v/v), and immediately analyzed and quantified by LC/MS/MS. Eicosanoids were separated by reverse-phase chromatography and measured using electrospray ionization in negative ion mode and multiple reaction monitoring. Eicosanoids were identified by matching their MRM signal and chromatographic retention time with those of pure identical standards.

**Total RNA and scRNA sequencing**

Total RNA was isolated from NTL tissues of *Scd2^f/f^;CC* vs. *Scd2^f/f^* mice which underwent the DEN + WAD regimen, using PicoPure RNA

**Fig. 4 | HCC growth is dependent on LTB4R2. a** IB analysis of LTB4R2 and NaKATPase (membrane), pYAP1, pGSK3β, GSK3β, HuR, GAPDH (cytosolic), YAP1 and CTNNB1 (nuclear) proteins from B6 mice subjected to the DEN + WAD regimen and injected with AAV vector ($4 \times 10^{11}$ GC per mouse) expressing *SCR-shRNA (SCR-sh)* vs. *LTB4R2-shRNA (LTB4R2-sh)* one month prior to the end of experiment (*n* = 6 mice per group). **b** Liver tumor development in the mice with *SCR-sh* vs. *LTB4R2-sh* treatment depicted by representative images and total tumor volume and multiplicity. *$p < 0.05$ vs. *SCR-sh* mice by two-sided t-test. Data presented as means ± SEM (*n* = 9 mice each). **c** qPCR data for oxylipin synthetic genes, *LTB4R2*, and *YAP1* in patient HCC (*n* = 6) vs. normal subject livers (*n* = 6). *$p < 0.05$, **$p < 0.01$ vs. normal liver by two-sided t-test. Data presented as means ± SEM. **d** Representative IHC-HRP staining for CYP1B1 and LTB4R2 of patient HCC liver sections (×200) from four

patient samples examined. Areas demarked by broken lines are HCC. **e** Top: Growth of patient HCC organoid (model-1 and model-2) in the presence of DMSO (vehicle) or the LTB4R2 antagonist LY255283. *$p < 0.05$, **$p < 0.01$, ***$p < 0.005$ vs. DMSO by two-way ANOVA with post hoc test. Data presented as means ± SEM (*n* = 3 experiments). Bottom: Growth of patient HCC organoid model-1 and model-2 without (CTRL) or with infection with adenovirus expressing *scrambled shRNA (shCTRL)* or *LTB4R2* shRNA (shLTB4R2). *$p < 0.05$, ***$p < 0.005$ vs. DMSO (*n* = 3 experiments). **f** Schematic diagram of HCC promotion initiated by SCD-CYP1B1-dependent release of 12-HHTrE by *Lart*⁺*Fbln2*⁺ aHSC, activating LTB4R2-CTNNB1-YAP1 pathway in HCC cell. For all relevant figures, source data and exact *p* values are provided in the Source Data file.

isolation kit (Thermo Fisher Scientific) and RNA integrity was verified by Experion analysis (Bio-Rad Laboratories). PolyA RNA was collected by using Illumina Truseq V2 polyA beads and sequencing was performed on a NextSeq 500 with V2 chemistry at the Molecular and Genomic Core of the USC Norris Cancer Center and sequencing data were analyzed by the Partek Flow software (Partek Inc). Gencode M3 was used to quantify the aligned reads to genes using Partek E/M method. The gene level aligned read counts were normalized for all sample using Upper Quartile normalization before subjected to differential expression analysis using Partek Gene Specific Analysis method. For scRNA-seq, the library was prepared immediately after cell sorting with the target cell number of 10,000 and validated on the Agilent TapeStation (Agilent Technologies) and quantified by using Qubit 2.0 Fluorometer (Invitrogen) and qPCR (KAPA Biosystems). The samples were sequenced using a $2 \times 150$ Paired End configuration on the Illumina HiSeq, and raw sequencing data were converted to fastq files and de-multiplexed using the 10x Genomics' Cell Ranger software version 3.1.0. Subsequently, UMI and cell barcode de-convolution and mapping to the mm10 reference genome were performed with the software to generate the final digital gene expression matrices and cloupe files using the Cell Ranger count command with default parameters. The cells which passed the quality assessment by a barcode rank graph, were subjected to sequencing and the data analysis was performed with 10X Genomics' Loupe Browser software.

## TEAD promoter-luciferase assay

Huh7 cells were transfected at 70% confluency in DMEM growth media containing 5% FBS with 8xGTIIC-Tead4-luciferase (Addgene), a 624b *YAP1* enhancer-luciferase construct with or without the mutations in the first intronic TCF site[7], and renilla-luciferase plasmids (Addgene) using Biolab transfection reagent. After overnight, the cells were treated with 12(S)-HHTrE (50 nM, cat#34590),12-HETE (100 nM, cat#34570), Ozagrel (1 μM, cat#70515), LY255283 (10 μM, cat#70715) obtained from Cayman Chemical Company and the YAP-TEAD binding inhibitor Super-TDU (1-31) TFA (300 nM, MedChemexpress, cat#HY-P1728A) with respective vehicle controls in serum-free medium for 24 h. Protein concentrations were determined in cell lysates and equal amounts of proteins were analyzed for firefly and renilla luciferase activities using Dual-Glo Luciferase Assay Kit (Promega).

## Chromatin immunoprecipitation

Enrichment of CTNNB1 at the proximal promoter region (YAP-Y2) and the first intronic enhancer region (YAP-Y3) of YAP1 gene were analyzed by ChIP-qPCR analysis as previously described[7]. Briefly, Huh7 cells ($8 \times 10^6$ cells) were incubated overnight with 12-HHTrE vs. ethanol vehicle and fixed in formalin (37%) for 10 min. Cross-linked chromatins were subjected to enzymatic digestion using ChIP-IT Express kit (Active motif, cat#53009) and analyzed on 1% agarose gel for fragment sizes of 700–200 bp. 3 μg of anti-β catenin (BD Transduction, 610154) and anti-RNAPII (Covance, 8WG16/MMS-12-6R) were added to 8 μg of chromatin for immunoprecipitation. The primers used for real-time PCR for YAP1-Y2 and YAP-Y3 regions were:

YAP-Y2-FW: 5′CAGAGGAAGGAAGAGCCGAGAGG3′
YAP-Y2-REV: 5′CGCCCGACTGAGACAGAAACT-3′
YAP-Y3-FW: 5′-GCGTGTTGGTTTCCCAGTTGTAGA-3′
YAP-Y3-REV: 5′-GCGCAACGTACAGATGTGGCTAAT-3′

## Cloning human *LRB4R2* promoter/intron deletion-luciferase constructs

UCSC Genome Browser was used to examine the distribution of potential regulatory elements within a human LTB4R2 proximal promoter/first intron region spanning from −1517 bp and extending upstream to +258 bp or +344 bp relative to TSS. To generate deletion-reporter constructs, the following different segments of the promoter with a desired restriction sequence was cloned in pGL3-Luciferase vector by PCR amplification of genomic DNA and ligation of amplicon: −224/+258, 492/+258, −1117/+258, −224/+344, −492/+344, −1117/+344, −1517/+344. Forward and reverse PCR primers used are as follows:

hLTB4R2-F-492-KpnI: CGG GGTACC GAACCTAGCACCATGC CTTAC

hLTB4R2-F-492-MluI: GAATTC ACGCGT GAACCTAGCACCATG CCTTAC

hLTB4R2-F-1117-KpnI: CGG GGTACC GGTAGAACAACTCTCT CTCAC

hLTB4R2-F-1117-MluI: GAATTC ACGCGT GGTAGAACAACTCTC TCTCAC

hLTB4R2-F-1517-KpnI: CGG GGTACC GACGTGACAGAGATGTG AATG

hLTB4R2-F-1517-MluI: GAATTC ACGCGT GACGTGACAGAGAT GTGAATG

hLTB4R2-F-224-KpnI: CGG GGTACC GATGAGAACAGAAGCA GGAC

hLTB4R2-F-224-MluI: GAATTC ACGCGT GATGAGAACAGAAGCAG GAC

hLTB4R2-R + 74-HindIII: CTAG AAGCTT GGAGAAGCTGAAACC TTCCGC

hLTB4R2-R + 258-HindIII: CTAG AAGCTT GTGGTAAACAGGCAT AAAGTC

hLTB4R2-R + 344-HindIII: CTAG AAGCTT CTCAAACACAACTCC TTCTTG

Resultant plasmids were amplified and used for transient transfection to determine the promoter activity in response to 12-HHTrE vs. EtOH in Huh7 cells using Dual-luciferase assay system as described above.

## RT-qPCR and immunoblot analysis

Total RNA was extracted from liver tissue or cells by the Quick-RNA MiniPrep (Zymo Research, Irvine, CA) and reverse-transcribed using high-capacity cDNA Reverse Transcription Kit (Thermo Fisher Scientific, Waltham, MA, USA). Real-time PCR was performed by amplifying cDNA for 40 cycles using primers shown in Supplementary Table 2 and the SYBER Green PCR master mix (Applied Biosystems, Foster City, CA) on an. Each threshold cycle (Ct) value of samples was normalized to the Ct value of the housekeeping genes (36B4) and subsequently to their

control samples. For immunoblotting, total liver proteins were extracted with the RIPA buffer (Santa Cruz Biotechnology), or nuclear, cytoplasmic, and membrane proteins were extracted from Huh7 cells and liver tissues using NE-PER™ extraction reagent (Thermo Fisher Scientific). 10 µg of proteins were mixed with a 6X SDS sample buffer, resolved on 8–10% SDS-polyacrylamide gel electrophoresis, transferred onto nitrocellulose membrane, and immunoblotted. The dilutions and sources of primary antibodies were listed in Supplementary Table 1. Raw images of all immunoblots are provided in Source Data file.

## Immunofluorescence microscopy and immunohistochemistry

IF staining was performed in mouse liver sections with 4% paraformaldehyde following antigen retrieval with citric acid buffer (pH 6.0) for 10 min at 98 °C. After blocking the endogenous peroxidase activity and non-specific protein binding sites, the sections were incubated with primary antibodies for rabbit-YAP (1:500) from Abcam, and YAP expressing cells were co-stained with mouse-α-SMA (1:500) from Sigma-Aldrich, goat-HNF4α (1:500) from Santa Cruz Biotechnology, mouse-SOX9 (1:1000) from Merck, followed by incubation with the fluorescent secondary antibody (Supplementary Table 1). Stained slides were examined under a fluorescence microscope (BZ9000; Keyence Corp., Osaka, Japan). Nuclei were stained using DAPI (4′6-diamidino-2-phenylindole; Sigma-Aldrich). Total numbers of stained cells were counted, and the values per unit of area were calculated using AxioVision Rel 4.9.1. For confocal 3D imaging, samples were treated with SeeDB44 overnight after staining. Images were acquired under a confocal microscope (FV-1000 or FV3000; Olympus) with a 30× silicone immersion lens (UPLSAPO30XS; Olympus). 3D images were reconstructed with IMARIS software (Bitplane, Zurich, Switzerland). Huh7 cells grown on coverslip were fixed for 30 min in 4% paraformaldehyde solution, washed, and permeabilized with 0.2% triton100-X in 1X PBS for 15 min at room temperature. The cells were then incubated with blocking buffer containing 5% FBS, 0.2% triton100-X in 1X PBS for 120 min at room temperature followed by overnight incubation with primary antibody of rabbit-YAP (1:100). Slides were washed in 1X PBS with 0.1% Tween-20 for three times and incubated with goat anti-rabbit-Alexa Fluor 594 secondary antibody (1:100) from Jackson ImmunoResearch Laboratory) in 1% BSA with 0.1% Tween-20 for 60 min at room temperature. Nuclei were stained with DAPI. Slides were visualized under Nikon microscope. Intensity of YAP + cells was calculated using NIS-Element Viewer Software. Paraffin-embedded untreated resected HCC patient samples were sectioned with 4 µm-thickness and adhered to charged glass slides (Superfrost Plus; Fisher Scientific, Waltham, MA, USA). Sections were incubated at 60 °C for 1 h prior to deparaffinization in xylene and then rehydrated in 100%, 95%, and double-distilled water, successively. Sections were heat-induced in retrieval buffer at pH 6.0, incubated with peroxidase block (Dako, Mississauga, ON, Canada) for 20 min followed by blocking (5% goat serum in 1× PBS-Tween20) for 1 h. Sections were then incubated overnight at 4 °C with primary antibodies diluted in blocking buffer. Primary antibodies used were: rabbit anti-CYP1B1 (Cytochrome P450 1B1: Boster Biological Technology, CA, PB9546; dilution 1:1000), rabbit anti-LTB4R2 (MyBioSource, Inc., CA, MBS243185; dilution 1:200). The detection system used was the EnVision+ System-HRP kit (Dako, K4007). Sections were counterstained with hematoxylin prior to dehydration and mounted with Permount (Fisher, SP-15-100). The first section of each series was stained with Hematoxylin and Eosin (H&E) for an initial histopathological assessment. Slides were scanned using the Aperio AT Turbo system (total magnification of 400×) and images were viewed using the Aperio ImageScope software program.

## Tumor organoid derivation, culture, and testing

Patient HCC organoid was prepared as previously described[45]. Briefly, liver tumors were minced and digested in sterile digestion media (PBS, 0.125 mg/mL collagenase from *clostridium histolyticum*, 0.125 mg/mL dispase II, and 0.1 mg/mL DNase I), strained through a 70 µm strainer and washed with basal media (Advanced DMEM/F-12, 1% glutamine, 1% penicillin/streptomycin, 10 mM HEPES). Cells were counted, washed, resuspended at 50,000 cells per 50 µL Matrigel (Corning), plated in 24 well plates, and cultured in murine tumor organoid media (basal media, 1:50 B27, 1 mM *N*-acetylcysteine, 10% Rspo1-conditioned media, 10 mM nicotinamide, 10 nM recombinant human [Leu[15]]-gastrin I, 50 ng/mL recombinant mouse EGF, 100 ng/mL recombinant human FGF10, and 50 ng/mL recombinant human HGF) until organoids formed. For passage, organoids were taken out of Matrigel in basal media, spun down at $300 \times g$ for 5 min at 4 °C, mechanically broken by passing through a 21-gauge needle, washed in basal media, and re-plated in Matrigel. Patient-derived organoids were cultured in human tumor organoid media (basal media, 1:50 B27 no vitamin A, 1:100 N2, 1 mM *N*-acetylcysteine, 10% Rspo1-conditioned media, 10 mM nicotinamide, 10 nM recombinant human [Leu[15]]-gastrin I, 50 ng/mL recombinant human EGF, 100 ng/mL recombinant human FGF10, 25 ng/mL recombinant human HGF, 10 µM forskolin, and 5 µM A83-01) and passaged as above. For assays in organoid lines, 96 well plates were first coated with a 50:50 solution of basal media:Matrigel (35 mL/well), which polymerized for 15 min at 37 °C. Tumor organoids were taken out of Matrigel, broken, and washed. Tumor organoids were seeded at 1000 cells per well and were treated the following day with an antagonist in technical triplicate. Final DMSO concentrations were kept below 0.5%. Organoid growth was measured with an XTT cell proliferation assay. Growth data were analyzed by normalizing individual antagonist-treated well values to DMSO-treated wells. For adenovirus infection of organoids, organoids were collected in Cell Recovery Solution (Corning) and were left to rotate at 4 °C for 1 h to dissolve the Matrigel. Organoids were spun down at $300 \times g$, 4 °C, for 5 min, dissociated into single cells with TrypLE (Gibco) by rotating at room temperature for 5 min, and cells were centrifuged at $300 \times g$, 4 °C, for 5 min. Cells were resuspended in transduction media (human tumor organoid media with 10 µM Y-27632 and 3 µM CHIR99021), and a corresponding amount of virus to achieve a MOI of 15. Cell suspensions were distributed into ultra-low attachment 24-well plates (Corning), and parafilm-wrapped plates were then spun at $600 \times g$ for 1 h at 32 °C. Following centrifugation, cells were incubated at 37 °C for 4−6 h. Infected cells were collected in 15 ml conical tubes, centrifuged at $300 \times g$ for 5 min at 4 °C, and redistributed into new 24-well plates in Matrigel + human tumor organoid media. Organoids were harvested on days 1 and 4 to confirm *LTB4R2* knockdown efficiency by qPCR.

## TCGA-LIHC cohort analysis

To assess the association of LTB4R2 mRNA expression of HCC patient survival, we used the publicly available TCGA-LIHC cohort data set of 358 primary tumor samples with Illumina HiSeq expression data (https://urldefense.com/v3/__https://xenabrowser.net/datapages/?dataset=TCGA.LIHC.sampleMap*2FHiSeqV2&host=https*3A*2F*2Ftcga.xenahubs.net&removeHub=https*3A*2F*2Fxena.treehouse.gi.ucsc.edu*3A443__;JSUlJSUlJSU!!LIr3w8kk_Xxm!oA-Ncipbp4sTfgx-peYujkrl8itm4ocffLGo5SaKPHOXvy_X9DPwGgTYJjaOpUve_iKR1hFjbPAsp6kePxU$) via the UCSC Xena platform (https://xenabrowser.net/datapages/?cohort=TCGA%20Liver%20Cancer%20(LIHC)&removeHub=https%3A%2F%2Fxena.treehouse.gi.ucsc.edu%3A443). Patients with (92 patients) or without (263 patients) *CTBBN1* mutations were stratified based on the TCGA-LIHC mutation data (https://xenabrowser.net/datapages/?dataset=mc3%2FLIHC_mc3.txt&host=https%3A%2F%2Ftcga.xenahubs.net&removeHub=https%3A%2F%2Fxena.treehouse.gi.ucsc.edu%3A443) and examined in each group to generate the Kaplan Meier survival plots for high vs. low LTB4R2 expression patients by referring to patients' survival data shown at https://xenabrowser.net/datapages/?dataset=survival%2FLIHC_survival.txt&host=https%3A%2F%2Ftcga.xenahubs.net&removeHub=https%3A%2F%2Fxena.treehouse.gi.ucsc.edu%3A443.

## Statistical analysis

All numerical data are expressed as the means ± SEM of replicate experiments. Statistical analysis for differences between two sets of data was analyzed by two-tailed *t*-test and for three or more different groups, by one-way ANOVA with post-hoc Tukey tests.

## Reporting summary

Further information on research design is available in the Nature Portfolio Reporting Summary linked to this article.

## Data availability

The mouse liver RNA-seq and HSC scRNA-seq data were deposited at the Gene Expression Omnibus with the accession code GSE193980 and GSE230843. We used publicly available scRNA-seq data from the NCI's Single-cell Atlas in Liver Cancer Data (scATlasLC: https://scatlaslc.ccr.cancer.gov/#/) and TCGA-LIHC data (https://pubmed.ncbi.nlm.nih.gov/28622513/) currently managed by the NCI's Genomic Data Common Data Portal (http://portal.gdc.cancer.gov/) as described in Methods. Source data are provided in this paper. The remaining data are available within the Article, Supplementary Information, or Source Data file. Source data are provided in this paper.

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

## Acknowledgements

The authors thank the Animal Core, Morphology Core, Administrative Core, Integrative Liver Cell Core (R24AA012885) of the Southern California Research Center for ALPD and Cirrhosis (P50AA011999), and the Liver Histology Core for USC Research Center for Liver Diseases (P30DK048522) for their animal experimental, histological, and administrative, cell isolation and culture services. FACS was performed by Flow Cytometry and Immune Monitoring Core of the USC Norris Comprehensive Cancer Center (P30CA014089), total RNA and scRNA sequencing was performed by Molecular Genomic Core of the USC Norris Comprehensive Cancer Center, and lipidomic analysis by UCSD Lipidomic Core. Liver tumor organoid experiments were facilitated by the services and facilities of the Tisch Cancer Institute supported by the NCI Cancer Center Support Grant (P30 CA196521). This work was supported by grants, U01AA027681 (M.K., H.T.), P50AA011999 (H.T.), and R24 AA012885 (H.T.); R01CA256480, R01CA249204, R01CA248984, and ISMMS seed fund to E.G.; and IK6BX004205 (H.T.: BLR&D Research Career Scientist Award) and I01 BX001991 (H.T.: VA Merit Review) from Department of Veterans Affairs. S.A.'s research scholarship was supported in part by the Lee Summer Research Fellowship of the Southern California Research Center for ALPD and Cirrhosis and Japan Student Services Association. Patient HCC tissues were supplied by the Norton Herrick Surgical Biorepository of the University of Southern California. Healthy human livers were provided by the Kansas University Liver Center of Kansas University Medical Center supported by COBRE grant numbers P20 GM103549 and P30 GM118247. Immunohistochemical and IF staining work by A.K.-L. and A.L. were supported by the McGill University Health Centre Foundation. A.R. was supported by an NCI training grant and NCI F32 fellowship (F32CA247414-01). We also thank the University of Tokyo IQB Olympus Bioimaging Center (TOBIC) for supporting confocal IF microscopy and image acquisition.

## Author contributions

H.T. proposed the original hypothesis, designed and supervised the study, performed scRNA-seq analysis, analyzed and interpreted the data, wrote and revised the manuscript. S.S. performed a majority in vitro and in vivo experiments, analyzed the data with H.T., produced figures and tables, and drafted the method section. S.A. performed the LTB4R2 antagonist experiment on Huh7 cells and assisted S.S. Y.N. and H.Y.C. performed IF analysis of the mouse model livers, R.S. performed IF analysis of Huh7 cells, and A.K.-L. performed IHC staining of patient HCC sections under the supervision of A.L. S.Q.P. isolated liver mesenchymal cells for FACS. G.Y. provided the YAP1 enhancer constructs. L.S. provided patient HCC tissues for qPCR analysis. S.P.M. provided liver tissues from Ctnnb1f/f;AlbCre vs. Ctnnb1f/f mice. Y.C. and M.L. performed bioinformatic analysis. K.M. cloned LTB4R2 promoter deletion-luciferase constructs. A.R. performed HCC organoid experiments under the supervision of E.G. M.K. made scientific and editorial suggestions and comments.

## Competing interests

The authors declare no competing interests.

## Additional information

[1]Southern California Research Center for ALPD and Cirrhosis, Keck School of Medicine of the University of Southern California, Los Angeles, CA 90033, USA. [2]Department of Pathology, Keck School of Medicine of the University of Southern California, Los Angeles, CA 90033, USA. [3]Laboratory of Cell Growth and Differentiation, Institute for Quantitative Biosciences, The University of Tokyo, Tokyo 113-0022, Japan. [4]Icahn School of Medicine at Mount Sinai Hess Center for Science and Medicine, New York, NY 10029, USA. [5]USC Libraries Bioinformatics Services of the University of Southern California, Los Angeles, CA 90089, USA. [6]Research Institute of the McGill University Health Centre, Montreal, QC H3A 0G4, Canada. [7]Department of Surgery, Pennsylvania State University, Hershey, PA 17033, USA. [8]Department of Surgery, Keck School of Medicine of the University of Southern California, Los Angeles, CA 90033, USA. [9]Department of Pathology, University of Pittsburg School of Medicine, Pittsburg, PA 15213, USA. [10]Department of Molecular Microbiology and Immunology, Keck School of Medicine of the University of Southern California, Los Angeles, CA 90033, USA. [11]Department of Pharmacology, University of California San Diego, La Jolla, CA 92093, USA. [12]Department of Veterans Affairs Greater Los Angeles Healthcare System, Los Angeles, CA 90073, USA.
✉e-mail: htsukamo@med.usc.edu

