## [Peer Review File · Nature Communications]

Hepatic stellate cell stearoyl co-A desaturase activates leukotriene B4 receptor 2- β -catenin cascade to promote liver tumorigenesisREVIEWER COMMENTS

Reviewer #1 (Remarks to the Author): with expertise in hepatocellular carcinoma

This is a follow-up study to the group's previous work published in *Gastroenterology* in 2017 – SCD promotes liver fibrosis and tumor development in mice via Wnt signaling and stabilization of low-density lipoprotein receptor-related proteins 5 and 6. They found SCD to be expressed from aHSCs and in HCC. In this study, they found ablation of SCD in aHSC to suppress beta-catenin and YAP1 in the tumor microenvironment and prevent HCC in mice. Mechanistically, they found SCD to promote 12-HHTrE, which will bind to LTB4R2 on HCC cells to regulate beta-catenin/YAP1 signaling. This is an interesting study. Authors utilized knockout mice, NGS including scRNA-seq as well as various cell/molecular biology techniques. I do have several concerns/suggestions.

1. Page 4 – Authors talk about Scd2f/f;CC vs. Scd2f/f mice without providing information to what these mice are. More information about these mice should be given up front.
2. This study talks about liver tumorigenesis but a lot of the results are not derived from a HCC mouse model. For instance, data presented in Figures 1c-f are all in a normal liver setting. Similarly, data to show “CTNNB1 regulation of hepatic YAP1 and SCD1 expression” is also obtained from Ctnnb1f/f;Albcre mice lacking CTNNB1 in hepatocytes. Is normal hepatocytes representative of the situation in HCC?
3. What is the mechanism by which aHSC SCD2 in control of LTB4R2 expression and its high-affinity oxylipin ligand 12-HHTrE? Is 12-HHTrE only controlled by CYP1B1? How does CYP1B1 control 12-HHTrE?
4. Figure 2b-c: What do the numbers 1 to 6 or 1 to 5 mean on top? Are these different clones? Different replicates?
5. Figure 2i: Is the data significant?
6. Is LTB4R2 specific to HCC cells?
7. Multiplex immunofluorescence is suggested to examine the localization of the proteins of interest to ensure that the findings are not just correlative.
8. One major limitation of this study is the lack of clinical relevance. HCC patient clinical data to show at least correlation of targets of interest is needed.

Reviewer #2 (Remarks to the Author): with expertise in liver lipidomics

This report extends a previous reported by the authors in which they described a novel Wnt/beta-catenin-stearoyl CoA desaturase (Scd)-LDLR-related protein (Lrp) positive loop and cellular cross talk that was involved in liver fibrosis and tumor formation. The new information in this report deals with the discovery of a metabolic link between Scd and an oxylipin (12-HHTrE) and Cyp1B1. This oxylipin is a mono-oxidized 17-carbon polyunsaturated fatty acid. This oxylipin had been previously described as a leukotriene B4 receptor (LTB4R2) agonist. Another key piece of information is the finding that activated hepatic stellate cells (aHSC) express cytochrome 1B1 (Cyp1b1), but no other enzymes associated with oxylipin metabolism. Cyp1b1 are known to oxidize many compounds including fatty acids; in this case it generates oxylipins. A final key piece of information addresses translational relevance. The authors report finding that Cyp1B1-expressing cells near LTB4R2-positive cancer cells in human HCC. Based on these results, the authors have extended their previous study by linking the Wnt/beta-catenin-Scd-Lrp5/6 to an oxylipin (12-HHTrE), and enzyme involved in oxylipin production (Cyp1b1); and they provide supporting evidence for this pathway may exist in human HCC tumors.

While much of the data reported supports the authors conclusions, this reviewer has concerns with the lipidomic analysis.

1. Line 7, p 6: “Scd2 is an enzyme essential for generation of MUFA, which serve as precursors for eicosanoids”. This statement is not entirely correct. While it is correct that Scds convert saturated fatty acids to monounsaturated fatty acids, it is not correct that MUFA are precursors to eicosanoids. First, 12-HHTrE is 17-carbon polyunsaturated fatty acid (PUFA). The source of the 17 PUFA is not

described or known. The oxidation at carbon 12 may be mediated by Cyp1B1. The formation of the other 2 double bonds, however, likely requires the activity of other enzymes, e.g., FADS1, FADS2. Alternately, the 17-carbon PUFA may have come from the diet.

12(S)-HHTrE, from Cayman Chemical Co

2. The display of the lipid species in "Extended data figure 1" needs revision. First, the figure legend states that red and blue arrows point to arachidonic acid metabolites. This is not correct. 9-Hode, 9,10-Epome are derived from linoleic acid (C18:2, n-6). In addition, the arrows obscure the error bars. There is no indication of statistically significant difference induced by a change in Ctnnb expression. There are other oxylipins (e.g. 5,6-EET) that appear more affected by changes in Ctnnb expression than 12-HHTrE. This raises doubts as to whether the 12-HHTrE is the key oxylipin involved in the Wnt/beta-catenin-Scd-Lrp5/6 to an oxylipin pathway. Interestingly, no omega-3 PUFA derived oxylipins were presented in this figure. Omega-3 PUFA and oxylipins derived from these PUFA have been suggested to be protective against hepatic inflammation and fibrosis in diet-induced fatty liver disease.

3. Finally, as stated in the text and described previously by others Scd2 is not expressed in humans. The two SCDs in humans are SCD1 and Scd5; and Scd5 is expressed in several hepatic cells, including stellate, cholangiocytes, endothelial and T-cells. (Human proteome atlas).

Reviewer #3 (Remarks to the Author): with expertise in hepatocellular carcinoma

Sinha, Aizawa et al show that Scd2 and Cyp1b1-mediated production of leukotriene B4 receptor 2 (LTB4R2) ligand 12-hydroxyheptadecatrienoic acid (12-HHTrE) promotes tumor growth via LTB4R2. This hypothesis is supported by in vivo experiments in mice with Col1a1Cre-mediated deletion of Scd2 and AAV8-TGB-shLRB4R2-mediated silencing of LTB4R2 in tumor cells as well as in vitro experiments in Huh7 tumor cells. The authors demonstrate that 12-HHTrE and LTB4R2 induce these effects through upregulation of b-catenin and YAP in tumor cells. While the manuscript contains a large amount of in vitro and in vivo data that generally support the hypothesis, there are also several gaps and weaknesses:

Major points

1. Previous studies have shown a role for a b-catenin-independent activation of YAP/TAZ via alternative Wnt signaling (Park et al, Cell . 2015 Aug 13;162(4):780-94). The authors should address this point. In particular the expression of YAP should by western blot is questionable as most YAP is expressed in ductular cells and a change in this compartment, which is also deleted by AlbCre and could be altered in response to CTNNB1 deletion in the liver, could be responsible for the results in the western blot. The authors have determined YAP by IHC, but the data do not look convincing: A. Most YAP is outside the HNF4a nuclei and a number of nuclei that barely have YAP are designated as YAP-expressing. B. The authors need to analyze this by confocal to make sure the signal is in the same cell and also display the red and green fluorescence channels separately so that this is clear. Importantly, since the authors state that the effects are on tumor cells, these analysis should be performed in tumor tissue and not in non-tumor liver.

2. The authors show alterations of YAP target genes and then begin focusing on YAP. However, YAP and TAZ are both responsible for the regulation of these genes and TAZ also has a key role in carcinogenesis, including HCC (Wang et al, J Hepatol. 2022 Jan;76(1):123-134.) The authors should therefore determine if Wwtr1 mRNA and protein are also regulated by the investigated pathways, i.e. Scd2-deletion in HSC and CTNNB1-deletion in hepatocytes.

3. The authors state in the abstract "...we report that selective ablation of 5 stearyl CoA desaturase (Scd) in aHSC globally suppressed nuclear β -catenin (CTNNB1) and YAP1 in the tumor microenvironment", and similar statements are made through the manuscript. This is not correct - the authors mix-up non-tumor liver and the TME (the TME are the non-malignant cells within a tumor - however, the CTNNB1 and YAP1 expression change in epithelial/tumor cells as shown in various experiments).

4. The author suggest a key role for 12-HHTrE from aHSC based on their analysis in Fig.1F. Since aHSC are only a small portion of cells in the liver and that the WAD does not induce profound fibrosis, the authors should demonstrate that aHSC indeed produce a large proportion of this metabolite in the liver. Along, it is not clear why deletion of CTNNB1 in hepatocytes reduces the same metabolites - the authors state "supporting the importance of CTNNB1 in regulating these oxylipins" but do not mention that in the case of Scd2, the knockout is in aHSC, whereas in case of CTNNB1, the knockout is in hepatocytes. This may suggest that these metabolites are either produced by hepatocytes (and that this is indirectly regulated by Scd2-expressing HSC) or that b-catenin-expressing hepatocytes regulate the production of 12-HHTrE. The authors need to follow up on this. Also, the analysis in Fig.1f shows lower levels of virtually all metabolites in the Cre+ mice - does this mean all eicosanoid metabolites are reduced? This can be statistically evaluated and would be a different interpretation from specific metabolites being selectively reduced. Finally, the authors show that LTB4R2 antagonist LY255283 inhibited Huh7 cell proliferation in monocultures. This again, this suggests that 2-HHTrE is produced by tumor cells themselves (please see also major point 4). In summary, the in vivo and in vitro data require clarification on which cells are responsible for the production of 12-HHTrE and possibly adjustment of the overall hypothesis (please see also comment on Cyp1b1 and Scd2 expression in major point 6).

5. How is LTB4R2 regulated by Scd2 and its products? The authors show that exposure with 12-HHTrE leads to LTB4R2 upregulation but not how. Is this YAP- or b-catenin-mediated?

6. The authors show in Figure 3e that virtually all VitA+HSC produce Scd2 but that only a very small portion of these cells produce Cyp1b1. On the background of major point 4, it would be extremely important to determine what cells in the liver express Cyp1b1 and Scd2 using scRNA-seq. Given that HSC are only about 5-10% of liver cells and that the portion of Cyp1b1 expressing cells is very low among these, it is possible that other cells in the liver express Cyp1b1 or Cyp1b1 + Scd2. The author may chose to do this analysis in the TME of their model or use publicly available scRNA-seq datasets from mice, and should also confirm SCD1 and CYP1B1 expression in human data (publicly available scRNA-seq and snRNA-seq datasets exist). Given that humans do not express SCD2, it is important to know whether SCD1 is enriched in human HSC. It is possible that there are species-specific differences and that the role of SCD2 is mouse-specific, with SCD1 expressed in other cells fulfilling the same functions in patients.

7. The authors use AAV8 to knockdown of LTB4R2sh one month before the sac date. Figure 4a shows knockdown in non-tumor liver - but the authors postulate that effects of aHSC are on the tumor (which makes sense given that AAV8 injection was only 1 month before sac - and that they also showed effects of the pathway in tumor cells, i.e. Huh7 cells). Please show the knockdown in tumor. Related to that, it is also not clear how efficient AAV8 is in transducing tumor cells.

8. The authors postulate a key role for LTB4R2 in HCC, activated by a signaling cascade triggered through Scd-expressing activated HSC. Given that there is a dedicated section "Translational relevance", it is suprising that an analysis of how LTB4R2 affects HCC is missing survival. This can be easily done in the TCGA or similar datasets and needs to be added.

9. The numbers of the plots (n=9) and the legends (n=6) do not match for Fig.4b. The authors should also add the liver body weight ratio as indirect and unbiased measure of tumor load. In addition to not matching number in Fig.4b, there are also other discrepancies, e.g. Fig.2d, which shows many more data points than the n=3-6 fields; Fig.2F, where the NT and Veh groups have less than n=8 samples.

This needs to be addressed and all figure should be changed to display all data points as dot plots.

Minor points

1. The pathway analysis in Extended Figure 1 needs to be properly displayed - this needs to be ranked by p-value and shown in an unbiased manner as announced by the authors "To unbiasedly search for the mechanisms...".
2. There are a number of typos, e.g. Page 9, please correct "phosphor-ERK1/2" to "phospho-ERK1/2", page 26 "via tail vain" to "via tail vein"
3. Abstract states "selective ablation of stearyl CoA desaturase (Scd) in aHSC" - this is not correct as Scd2 was ablated.
3. Figure 3b lacks replicates and statistical analysis. This is needed to make a statement or data should be omitted.
4. The scRNA-seq analysis in Fig.3e lacks the tSNE plot for Scd2 in VitA- HSC.
5. There are many figures with 3 groups and statistical analysis showing specific comparison - but the posthoc ANOVA test applied is not mentioned in the methods. This needs to be addressed.
6. The methods need to describe the genetic background/backcrosses of mice, breeding scheme (littermates?, cohousing?) and gender. The WAD regimen needs to be described.
7. Efficiency of Scd2 knockout in HSC needs to be stated, as WAD may not activate all HSCs sufficiently to expressed Col1a1.

Reviewer #4 (Remarks to the Author): with expertise in liver, scRNAseq

In this manuscript, 'Hepatic stellate cell-derived oxylipin activates leukotriene B4 receptor 2- β -catenin-YAP1 cascade to promote liver tumorigenesis' Sinha et al., discovered hepatic stellate cell orchestrated 12-HHTrE-LTB4R2-CTNNB-YAP1 signalling axis in HCC and its potential therapeutic implications. Overall, this is exciting work finding the novel molecular interactions between the tumour and its microenvironment in HCC. Authors have employed elegant fate-mapping experiments to demonstrate the role of SCD2 mediated repression of YAP1 in normal liver tissues. Next, they showed the role of Wnt signalling in YAP1 and SCD expression regulation. Further, they linked SCD expression with LTB4R2 and 12-HHTrE expression and the implication of 12-HHTrE on YAP1 via LTB4R2 and CRNNB1. The authors established that hepatic stellate cells are the primary source of 12-HHTrE expression in CYP1B1 dependent manner. Thereby, the authors have found a comprehensive axis of 12-HHTrE-LTB4R2-CTNNB-YAP1 signalling in liver cancer. However, I have the following suggestions before this manuscript is accepted for publication.

Although authors have done scRNA-seq analysis to however, its done on sorted population. Can the authors perform unbiased scRNA-seq to show that hepatic stellate cells are the primary source of 12-HHTrE expression?

These are very interesting findings but its unclear if the same axis play an important role in human cancers. Can authors use human organoids or TCGA data to validate their findings in human tumors?

Minor comments:

The authors should provide full-length WB for the images.

Point-by-point responses to the reviewers' comments

Reviewer 1:

1. "Page 4 – Authors talk about - line 6-7, page 4 without providing information to what these mice are. More information about these mice should be given up front"

Response: More information on *Scd2f/f;CC* vs. *Scd2f/f* mice is now provided in the introduction – **line 5-6, page 4** and in Supplementary Methods. Additionally, the rationale and validity of the use of *Scd2f/f;CC* mice are discussed in **2nd paragraph of page 4 to page 5**.

2. "This study talks about liver tumorigenesis but a lot of the results are not derived from a HCC mouse model. For instance, data presented in Figures 1c-f are all in a normal liver setting. Similarly, data to show "CTNNB1 regulation of hepatic YAP1 and SCD1 expression" is also obtained from *Ctnnb1^{ff};Albcre* mice lacking CTNNB1 in hepatocytes. Is normal hepatocytes representative of the situation in HCC?"

Response: As suggested, we now include liver tumor cell analysis by confocal 3-D imaging analysis of nuclear YAP1 in *Scd2f/f;CC* vs. *Scd2f/f* mice (**Fig. 1e and f**). Further, qPCR data of tumor for *Yap1*, *Wwtr1*, and *Ltb4r2* (**Ext Data Fig. 1h**) are now included to show they are all coordinately reduced in tumors as observed in tumor-adjacent liver (TAL) without visible tumors but with microscopic tumor cells. We have used *Ctnnb1^{ff};AlbCre* mice lacking CTNNB1 in hepatocytes just to address if CTNNB1 positively regulates YAP1/TAZ and SCD-HUR-LRP6 pathway even in normal hepatocytes (**2nd paragraph of page 6**). The reviewer is correct that our study focus should be placed on liver tumors. For this reason other data on normal liver from *Ctnnb1^{ff};AlbCre* mice such as lipidomic results have been removed.

3. "What is the mechanism by which aHSC SCD2 in control of LTB4R2 expression and its high-affinity oxylipin ligand 12-HHTrE? Is 12-HHTrE only controlled by CYP1B1? How does CYP1B1 control 12-HHTrE?"

Response: These are very important mechanistic questions. For LTB4R2 regulation, our original data showed 12-HHTrE-mediated positive forward regulation of LTB4R2 is pre-translational (**Ext Data Fig. 2g and j**). We now show this regulation is prevented by CTNNB1 knockdown (KD) (**Ext Data Fig. 2m**). More mechanistically, our ChIP-seq bioinformatic search identified potential CTNNB1 enrichments in the LTB4R2 proximal promoter and first intron. Based on this information, we have cloned this region (-1517/+344) and their deletion segments from genomic DNA into the p-GL3-luciferase vector. Using these deletion-reporter vectors in transient transfection assay, we have shown -492/-224 and +258/+344 regions are responsible for 12-HHTrE-induced activation (**Fig. 2j**), in a manner dependent on CTNNB1 (**Ext Data Fig. 2n**), most likely leading to mRNA upregulation. Precise molecular mechanisms of CTNNB1-mediated regulation of these two regulatory regions need to be investigated by our future studies.

As for CYP1B1-mediated 12-HHTrE production, we added new evidence that CYP1B1 KD reduces 12-HHTrE release by LX2 cells and blunted CM-induced TEAD promoter activity (**Fig. 3i and j**). Since SCD positively regulates CTNNB1 in aHSC as we have previously shown, our data suggest the SCD-CTNNB1-CYP1B1-12-HHTrE pathway. How CTNNB1 regulates CYP1B1 is an aim of our future study but may involve CTNNB1-AhR dependent regulation of CYP1B1 transcription as described for other CYP. These discussions are now included in the **latter half of page 24**.

4. "Figure 2b-c: What do the numbers 1 to 6 or 1 to 5 mean on top? Are these different clones? Different replicates?"

Response: The numbers shown above the lanes are replicates of shRNA-KD experiments and this is now explained in the figure legend.

5. *“Figure 2i: Is the data significant?”*

Response: Yes, the enrichment of CTNNB1 and RNA POLII are significantly increased ($p < 0.05$) and this is indicated by * in the figure and its legend in the revision.

6. *“Is LTB4R2 specific to HCC cells?”*

Response: LTB4R2 is known to be expressed by different tissues (skin, intestine, spleen) and cell types including immune cells, mast cells, and eosinophils. However, in the context of HCC, we believe tumor cell expression of LTB4R2 is germane to the pathway described. We now show *Ltb4r2* expression is suppressed in tumors of *Scd2^{ff};CC* vs. *Scd2^{ff}* mice (**Ext Data Fig. 1h**) and in tumors of the wild type mice injected with AAV8-*Ltb4r2* shRNA vs. AAV8-SCR shRNA (**Ext Data Fig. 4c**), the technique which we now show to transduce both hepatocytes and tumor cells (**Ext Data Fig. a-b**).

7. *“Multiplex immunofluorescence is suggested to examine the localization of the proteins of interest to ensure that the findings are not just correlative.”*

Response: As described above, we included confocal dual IF 3-D imaging for nuclear HNF4A and YAP1 to demonstrate nuclear YAP1 is reduced in HNF4A+ liver tumor cells in *Scd2^{ff};CC* mice (**Fig. 1e-f and Ext Data Fig. 1g**).

8. *“One major limitation of this study is the lack of clinical relevance. HCC patient clinical data to show at least correlation of targets of interest is needed.”*

Response: As recommended, we have analyzed patient survival in relation to *LTB4R2* expression by using the TCGA-LIHC (the Cancer Genome Atlas Liver Hepatocellular Carcinoma) cohort data. As our results suggested the role of CTNNB1 in YAP1 and LTB4R2 regulation and the *CTNNB1* missense mutation is common in HCC, we stratified the patients with or without the *CTNNB1* mutation. However, in either patient group with (92 patients) or without (263 patients) the mutation, no significant difference in survival between patients with high vs. low *LTB4R2* expression was observed (**Ext Data Fig. 4e**). This obviously was rather disappointing. However, another translational study we performed with patient HCC organoids as suggested by another reviewer, clearly showed a growth suppressive effect of LTB4R2 KD or antagonism particularly with the organoid having higher LTB4R2 expression (**Fig. 4e**). We also showed by IHC of patient HCC sections, LTB4R2-expressing HCC cells surrounded by CYP1B1+ CAFs (**Fig. 4d**) and significantly increased *LTB4R2* mRNA expression in HCC vs. normal liver by qPCR (**Fig. 4c**). Thus, we believe the lack of the positive correlation between *LTB4R2* expression and poor survival in the TCGA HCC cohort, may have different explanations including non-quantitative nature of RNA-seq data used for the TCGA gene expression profiling. In fact, our RNA-seq analysis of the mouse model also failed to show a significant repression of *Ltb4r2* in *Scd2^{ff};CC* mice (**Suppl Table 1**) despite the repression detected by immunoblotting or qPCR analysis (**Fig. 1i and Ext Data Fig. 1h**). Other explanations may include focal LTB4R2 expression by actively growing HCC cells and stage-dependent LTB4R2 expression. These discussions are now included on **page 24-25**.

Reviewer 2:

1. *“Line 7, p 6: “Scd2 is an enzyme essential for generation of MUFA, which serve as precursors for eicosanoids”. This statement is not entirely correct. While it is correct that Scds convert saturated fatty acids to monounsaturated fatty acids, it is not correct that MUFA are precursors to eicosanoids. First, 12-HHTrE is 17-carbon polyunsaturated fatty acid (PUFA). The source of the 17 PUFA is not described or known. The oxidation at carbon 12 may be mediated by Cyp1B1. The formation of the other 2 double bonds, however, likely requires the activity of other enzymes, e.g., FADS1, FADS2. Alternately, the 17-carbon PUFA may have come for the diet.”*

Response: As pointed out by the reviewer, corrections were made to state that “SCD2 is an enzyme essential for generation of MUFAs which give rise to polyunsaturated fatty acids (PUFA) via elongation and desaturation. We therefore performed a lipidomic analysis for PUFA metabolites in *Scd2^{ff}*;CC vs. *Scd2^{ff}* livers”. We also revised to state “12-hydroxyheptadecatrienoic acid (12-HHTrE), a 17-carbon PUFA of ill-defined sources; 12-hydroxyeicosatetraenoic acid (12-HETE) a 20-carbon PUFA derived from arachidonic acid; 9- and 13-hydroxyoctadecadienoic acid (9-HODE and 13-HODE) and 9,10-epoxyoctadecenoic acid (9,10-EpOME) derived from linoleic acid. - **the last paragraph of page 6 to page 7.**

Cayman Chemicals has been corrected to Cayman Chemical Company.

2. “The display of the lipid species in “Extended data figure 1” needs revision. First, the figure legend states that red and blue arrows point to arachidonic acid metabolites. This is not correct.”

Response: Yes, the figure legend statement was incorrect. Correct statements have been made for these PUFA metabolites as described above. Old Ext Data Fig. 1d for lipidomic data from *Ctnnb1^{ff}*;AlbCre vs. AlbCre mice has been removed due to insufficient sample numbers and its minimal relevance to liver tumorigenesis as pointed out by other reviewers.

3. “Finally, as stated in the text and described previously by others *Scd2* is not expressed in humans. The two SCDs in humans are SCD1 and *Scd5*; and *Scd5* is expressed in several hepatic cells, including stellate, cholangiocytes, endothelial and T-cells. (Human proteome atlas).”

Response: Thank you for this comment. We performed qPCR analysis on the human HSC line LX2 cells, the cell model we used for SCD1 KD and CYP1B1 KD experiments to assess HSC-liver cancer cell crosstalk for the present study where this isoform issue may become important. This analysis showed LX2 cells express ~20-fold less SCD5 compared to SCD1 – **line 5-6, page 15.**

Reviewer 3:

1. “Previous studies have shown a role for a b-catenin-independent activation of YAP/TAZ via alternative Wnt signaling (Park et al, Cell . 2015 Aug 13;162(4):780-94). The authors should address this point. In particular the expression of YAP should by western blot is questionable as most YAP is expressed in ductular cells and a change in this compartment, which is also deleted by AlbCre and could be altered in response to CTNNB1 deletion in the liver, could be responsible for the results in the western blot. The authors have determined YAP by IHC, but the data do not look convincing: A. Most YAP is outside the HNF4a nuclei and a number of nuclei that barely have YAP are designated as YAP-expressing. B. The authors need to analyze this by confocal to make sure the signal is in the same cell and also display the red and green fluorescence channels separately so that this is clear. Importantly, since the authors state that the effects are on tumor cells, these analysis should be performed in tumor tissue and not in non-tumor liver.”

Response: The first point has been addressed as suggested. This alternative pathway is expected to phospho-inhibits LATS, reduces p(S127)YAP1 and increase nuclear YAP1. However, in LTB4R2 KD cells, we observed reduced nuclear YAP1 and no change in cytosolic p(S127)YAP1. Conversely, in 12-HHTrE-treated cells, increased nuclear YAP was not associated with reduced p(S127)YAP1. Thus, our results suggest LTB4R2-mediated increase in nuclear YAP is independent of posttranslational YAP1 regulation by LATS which can be mediated by alternative Wnt signaling. This discussion and the publication on YAP/TAZ regulation by alternative Wnt pathway, have been added in the revision – **line 9-12, page 10.**

Nuclear YAP1 positivity was assessed in HNF4A+ hepatocytes, ACTA2+ aHSC, and SOX9+ ductular cells by dual IF microscopy and morphometry as shown in **Fig. c-d.**

As suggested by the reviewer and briefly described above for the response to the Reviewer-1, we performed confocal 3D imaging of HNF4A and YAP1 IF staining of non-tumor and tumor areas as shown in a new **Fig. 1e and f** and **Ext Data Fig. 1g.** Our morphometric analysis based on this technique showed a significant reduction in nuclear YAP1 in tumor cells. Additionally, qPCR

analysis of tumor tissues confirmed repressed *Yap1*, *Wwtr1*, and *Ltb4r2* in *Scd2^{ff};CC* vs. *Scd2^{ff}* mice (**Ext Data Fig. 1h**).

2. “The authors show alterations of YAP target genes and then begin focusing on YAP. However, YAP and TAZ are both responsible for the regulation of these genes and TAZ also has a key role in carcinogenesis, including HCC (Wang et al, J Hepatol. 2022 Jan;76(1):123-134.) The authors should therefore determine if *Wwtr1* mRNA and protein are also regulated by the investigated pathways, i.e. *Scd2*-deletion in HSC and *CTNNB1*-deletion in hepatocytes.”

Response: As suggested, we determined *Wwtr1* mRNA and TAZ protein in TAL and tumors and showed coordinated regulation as YAP1 - **Ext Data Fig. 1e, f, h, i, k**.

3. “The authors state in the abstract “...we report that selective ablation of 5 stearyl CoA desaturase (*Scd*) in aHSC globally suppressed nuclear β -catenin (*CTNNB1*) and YAP1 in the tumor microenvironment”, and similar statements are made through the manuscript. This is not correct - the authors mix-up non-tumor liver and the TME (the TME are the non-malignant cells within a tumor - however, the *CTNNB1* and YAP1 expression change in epithelial/tumor cells as shown in various experiments).”

Response: Thank you for pointing out the inappropriate use of TME. We have replaced TME with tumor-adjacent liver (TAL) throughout the revision and used TME when referring to non-malignant cells within tumors. We also replaced the term “non-tumor liver (NTL)” by TAL as NTL caused a confusion for the absence of tumor cells. This TAL tissue collected from the area right adjacent to liver tumor actually contained microscopic tumors although visible tumors were not evident as shown by **Fig. 1e**. We rationalized that the use of these TAL is ideal for studying aHSC/TME-tumor cell crosstalk as shown in **Fig. 1c-f**. This discussion is now included in the revision – **2nd paragraph, page 5**.

4. “The author suggest a key role for 12-HHTrE from aHSC based on their analysis in Fig. 1F. Since aHSC are only a small portion of cells in the liver and that the WAD does not induce profound fibrosis, the authors should demonstrate that aHSC indeed produce a large proportion of this metabolite in the liver.”

Response: This is an important question raised by the reviewer. As the reviewer pointed out, aHSC or a specific subpopulation of aHSC (*Lrat+Fbln2+* aHSC) which express CYP1B1, reside in the tumor-bearing liver in a relatively small number and whether 12-HHTrE produced by this small subpopulation quantitatively contributes to the concentration of the oxylipin in the liver is a logical question. To directly test this question, one has to conditionally ablate *Cyp1b1* in the *Lrat+Fbln2+* subpopulation in mice undergoing the DEN+WAD regimen to determine its effect on liver 12-HHTrE concentration. Unfortunately, we consider this technically time-requiring study to be beyond the scope of the current study. However, the present study showed that conditional ablation of *Scd2* in aHSC as validated by our new additional data (**Ext Data Fig 1a and d**), significantly reduced liver 12-HHTrE concentration (**Fig. 1h**), supporting the role of aHSC SCD2. This does not mean the direct role of aHSC in the overall oxylipin generation as aHSC might have regulated the oxylipin generation by other cell types (e.g., TAM) in a SCD2-dependent manner or by tumor cells as suggested by our *in vitro* experiment results on the paracrine-autocrine coupling (**last 2 sentences on page 15**). Therefore, a limited quantity of 12-HHTrE produced by aHSC may have a larger impact on the total liver oxylipin concentration via these paracrine cellular crosstalk effects. We incorporated this discussion in the revision – **2nd paragraph, page 25**.

“Along, it is not clear why deletion of *CTNNB1* in hepatocytes reduces the same metabolites - the authors state “supporting the importance of *CTNNB1* in regulating these oxylipins” but do not mention that in the case of *Scd2*, the knockout is in aHSC, whereas in case of *CTNNB1*, the knockout is in hepatocytes. This may suggest that these metabolites are either produced by

hepatocytes (and that this is indirectly regulated by *Scd2*-expressing HSC) or that *b*-catenin-expressing hepatocytes regulate the production of 12-HHTrE. The authors need to follow up on this. Also, the analysis in Fig. 1f shows lower levels of virtually all metabolites in the *Cre*⁺ mice - does this mean all eicosanoid metabolites are reduced? This can be statistically evaluated and would be a different interpretation from specific metabolites being selectively reduced.”

Response: Lipidomic data in *Ctnnb1*^{ff}; *AlbCre* mice have been removed as these data are not statistically conclusive and irrelevant to the study’s main focus on liver tumorigenesis as also commented by Reviewer-1. We have retained the IB and qPCR data on YAP1, TAZ, HUR, LPR6 to show CTNNB-SCD and CTNB1-YAP1 regulations appear present even in normal hepatocytes – **2nd paragraph of page 6.**

“Finally, the authors show that LTB4R2 antagonist LY255283 inhibited Huh7 cell proliferation in monocultures. This again, this suggests that 2-HHTrE is produced by tumor cells themselves (please see also major point 4). In summary, the *in vivo* and *in vitro* data require clarification on which cells are responsible for the production of 12-HHTrE and possibly adjustment of the overall hypothesis (please see also comment on *Cyp1b1* and *Scd2* expression in major point 6).”

Response: Discussed above under the responses to the first comment and the major comment #6 below.

5. “How is LTB4R2 regulated by *Scd2* and its products? The authors show that exposure with 12-HHTrE leads to LTB4R2 upregulation but not how. Is this YAP- or *b*-catenin-mediated?”

Response: As also described for other reviewer above, our original data showed 12-HHTrE-mediated positive forward regulation of LTB4R2 is pre-translational (**Ext Data Fig. 2g and j**). We now show this regulation is prevented by CTNNB1 knockdown (KD) (**Ext Data Fig. 2m**). More mechanistically, our ChIP-seq bioinformatic search identified potential CTNNB1 enrichments in the LTB4R2 proximal promoter and first intron. Based on this information, we have cloned this region (-1517/+344) and their deletion segments from genomic DNA into the p-GL3-luciferase vector. Using these deletion-reporter vectors in transient transfection assay, we have shown -492/-224 and +258/+344 regions are responsible for 12-HHTrE-induced activation (**Fig. 2j**), in a manner dependent on CTNNB1 (**Ext Data Fig. 2n**), most likely leading to mRNA upregulation. Precise molecular mechanisms of CTNNB1-mediated regulation of these two regulatory regions need to be investigated by our future studies.

6. “The authors show in Figure 3e that virtually all *VitA*+HSC produce *Scd2* but that only a very small portion of these cells produce *Cyp1b1*. On the background of major point 4, it would be extremely important to determine what cells in the liver express *Cyp1b1* and *Scd2* using scRNA-seq. Given that HSC are only about 5-10% of liver cells and that the portion of *Cyp1b1* expressing cells is very low among these, it is possible that other cells in the liver express *Cyp1b1* or *Cyp1b1* + *Scd2*. The author may chose to do this analysis in the TME of their model or use publicly available scRNA-seq datasets from mice, and should also confirm SCD1 and CYP1B1 expression in human data (publicly available scRNA-seq and snRNA-seq datasets exist). Given that humans do not express SCD2, it is important to know whether SCD1 is enriched in human HSC. It is possible that there are species-specific differences and that the role of SCD2 is mouse-specific, with SCD1 expressed in other cells fulfilling the same functions in patients.”

Response: Thank you for raising these important questions. To assess *Scd2* expression at the cellular level and to validate the use of *Scd2*^{ff}; *Col1a1Cre* (*Scd2*^{ff}; CC) mice for targeting aHSC, we performed scRNA-seq analysis of liver cells isolated from a mouse subjected to the liver tumor regimen of diethyl nitrosamine injection (DEN) and Western alcohol diet feeding (WAD). Among different cell type clusters identified, *Scd2* was expressed by *Lyve1*⁺ endothelial cells, *Adgre1*(F4/80)⁺ macrophages, *Lrat*⁺ HSC, *Fbln2*⁺ portal fibroblasts (PF), and *Itgax* (Cd11c)⁺ dendritic cells. However, *Col1a1* was selectively expressed by *Lrat*⁺ HSC and *Fbln2*⁺ PF

(**Extended Data Fig. 1b**). Thus, *Scd2* ablation mediated by *Col1a1* promoter-induced Cre should selectively occur in these two cell types in *Scd2^{ff};CC* mice. Further, we showed ~60% of *Fbln2⁺* cells were *Lrat⁺* aHSC in the DEN+WAD liver (**Ext Data Fig. 1c**). Collectively, these results support that *Scd2* ablation is sufficiently selective for aHSC in *Scd2^{ff};CC* mice.

As suggested, we also analyzed *Cyp1b1* expression in the same cell type clusters as discussed above and showed dominant *Cyp1b1* expression by *Fbln2⁺* cells (**Fig. 3c**). Further scRNA-seq of *Col1a1*-GFP⁺ cells revealed these *Fbln2⁺* cells are in large *Lrat⁺Fbln2⁺* aHSC (**Fig. 3e-g**).

To extend our *Cyp1b1* expression results to human HCC, we also re-analyzed the human HCC scRNA-seq data available from NCI's Single-cell Atlas in Liver Cancer (scAtlasLC:

<https://scatlaslc.ccr.cancer.gov/#/>) as suggested. Among different cell types in the human HCC TME, *CYP1B1* was expressed by CAF (cancer-associated fibroblasts), TAM (tumor-associated macrophages), and TEC (tumor-associated endothelial cells). *TBXAS1* was expressed by TAM and T cells but not in CAF (**Ext Data Fig 3i**). These results suggested that TAM may be a major source of the oxylipin in human HCC TME and *CYP1B1* may be a main source of oxylipin in CAF as shown in our mouse model (**Fig 3c and e**). Further, co-expression analysis showed, *CYP1B1*-expressing CAF are mostly *FBLN2*-positive (**Ext Data Fig 3j**) confirming our mouse scRNA-seq data (**Fig. 3f-g**). Based on these new data and our finding of the reduced 12-HHTrE and *Cyp1b1* in *Scd2^{ff};CC* mice, we have incorporated additional discussion that CAF/aHSC may release a mediator in a SCD-dependent manner which positive regulate the oxylipin release by other TME cells such as TAM (**2nd paragraph, page 25**). As SCD in aHSC potentially mediates this indirect crosstalk effect on other TME cell types, we have modified the manuscript title to "Hepatic stellate cell stearyl co-A desaturase activates leukotriene B4 receptor 2-β-catenin-YAP1 cascade.." from "Hepatic stellate cell-derived oxylipin activates leukotriene B4 receptor 2-β-catenin-YAP1 cascade.."

7. *"The authors use AAV8 to knockdown of LTB4R2sh one month before the sac date. Figure 4a shows knockdown in non-tumor liver - but the authors postulate that effects of aHSC are on the tumor (which makes sense given that AAV8 injection was only 1 month before sac - and that they also showed effects of the pathway in tumor cells, i.e. Huh7 cells). Please show the knockdown in tumor. Related to that, it is also not clear how efficient AAV8 is in transducing tumor cells."*

Response: Our additional result on the AAV8-mediated reporter GFP expression shows the evidence of the transduction in AFP+ tumor cells (**Ext Data Fig 4b**). We also show now a significant repression of *Ltb4r2* mRNA in tumors by this KD approach (**Ext Data Fig 4c**).

8. *"The authors postulate a key role for LTB4R2 in HCC, activated by a signaling cascade triggered through Scd-expressing activated HSC. Given that there is a dedicated section "Translational relevance", it is suprising that an analysis of how LTB4R2 affects HCC is missing survival. This can be easily done in the TCGA or similar datasets and needs to be added."*

Response: As also described above for other reviewers, we have analyzed patient survival in relation to *LTB4R2* expression by using the TCGA-LIHC (the Cancer Genome Atlas Liver Hepatocellular Carcinoma) cohort data. As our results suggested the role of *CTNNB1* in *YAP1* and *LTB4R2* regulation and the *CTNNB1* missense mutation is common in HCC, we stratified the patients with or without the *CTNNB1* mutation. However, in either patient group with (92 patients) or without (263 patients) the mutation, no significant difference in survival between patients with high vs. low *LTB4R2* expression was observed (**Ext Data Fig. 4e**). This obviously was rather disappointing. However, another translational study we performed with patient HCC organoids as suggested by another reviewer, clearly showed a growth suppressive effect of *LTB4R2* KD or antagonism particularly with the organoid having higher *LTB4R2* expression (**Fig. 4e**). We also showed by IHC of patient HCC sections, *LTB4R2*-expressing HCC cells surrounded by *CYP1B1*+ CAFs (**Fig. 4d**) and significantly increased *LTB4R2* mRNA expression in HCC vs. normal liver by qPCR (**Fig. 4c**). Thus, we believe the lack of the positive correlation between *LTB4R2* expression

and poor survival in the TCGA HCC cohort, may have different explanations including non-quantitative nature of RNA-seq data used for the TCGA gene expression profiling. In fact, our RNA-seq analysis of the mouse model also failed to show a significant repression of *Ltb4r2* in *Scd2^{ff}*;CC mice (**Suppl Table 1**) despite the repression detected by immunoblotting or qPCR analysis (**Fig. 1i and Ext Data Fig. 1h**). Other explanations may include focal LTB4R2 expression by actively growing HCC cells and stage-dependent LTB4R2 expression. These discussions are now included on **page 24-25**.

9. *"The numbers of the plots (n=9) and the legends (n=6) do not match for Fig.4b. The authors should also add the liver body weight ratio as indirect and unbiased measure of tumor load. In addition to not matching number in Fig.4b, there are also other discrepancies, e.g. Fig.2d, which shows many more data points than the n=3-6 fields; Fig.2F, where the NT and Veh groups have less than n=8 samples. This needs to be addressed and all figure should be changed to display all data points as dot plots."*

Response: Inconsistent data point numbers have been corrected in figure legends. Additional data on liver weight/body weight ratio have been added as **Ext Data Fig 4d**. Dot plots have been used for most of the figures presented.

Minor points:

1. *"The pathway analysis in Extended Figure 1 needs to be properly displayed - this needs to be ranked by p-value and shown in an unbiased manner as announced by the authors "To unbiasedly search for the mechanisms..."."*

Response: Ext Data Fig 1a has been replaced by **Supp Table 2** showing a summary of significantly downregulated genes in key pathways of relevance with p values.

2. *"There are a number of typos, e.g. Page 9, please correct "phosphor-ERK1/2" to "phospho-ERK1/2", page 26 "via tail vain" to "via tail vein"*

"Phosor-ERK1/2" and "tail vail" have been corrected.

3. *"Abstract states "selective ablation of stearyl CoA desaturase (Scd) in aHSC" - this is not correct as Scd2 was ablated"*

Response: "selective ablation of stearyl CoA desaturase (Scd)" has been corrected to *Scd2* ablation in Abstract.

4. *"Figure 3b lacks replicates and statistical analysis. This is needed to make a statement or data should be omitted."*

Response: **Old Fig 3b** has been omitted as suggested.

5. *"The scRNA-seq analysis in Fig.3e lacks the tSNE plot for Scd2 in VitA- HSC."*

Response: The t-SNE plots for *Scd2* in VitA- HSC have been added to **Fig. 3e**.

6. *"There are many figures with 3 groups and statistical analysis showing specific comparison - but the posthoc ANOVA test applied is not mentioned in the methods. This needs to be addressed."*

Response: ANOVA with posthoc test is now described in the methods and has been applied for **Fig. 1g, Ext Data Fig 1f, Fig. 2d-f, Ext Data Fig. 2l and m, Fig. 3a, h-j, Ext Data Fig, 3c-e**.

7. *"The methods need to describe the genetic background/backcrosses of mice, breeding scheme (littermates?, cohousing?) and gender. The WAD regimen needs to be described."*

Response: genetic background, breeding scheme, and gender for the genetic mice, the WAD regimen are now described in the Methods.

8. *"Efficiency of Scd2 knockout in HSC needs to be stated, as WAD may not activate all HSCs sufficiently to expressed Col1a1."*

Response: This has been tested by analyzing *Scd2* expression in total HSC and this additional result (**Ext Data Fig 1d**) showing ~95% repression.

Reviewer 4:

1. “Although authors have done scRNA-seq analysis to however, its done on sorted population. Can the authors perform unbiased scRNA-seq to show that hepatic stellate cells are the primary source of 12-HHTrE expression?”

Response: This has been performed for *Scd2*, *Col1a1*, and *Cyp1b1* for the mouse model as described above for the response to Reviewer 3 comment #6 – **Ext Data Fig 1a-b, Fig 3c**. Further, the data from re-analysis of publicly available human HCC scRNA-seq data, have been added in **Ext Data Fig 3i** as also describe above.

2. “These are very interesting findings but it’s unclear if the same axis play an important role in human cancers. Can authors use human organoids or TCGA data to validate their findings in human tumors?”

Response: We have analyzed patient survival in relation to *LTB4R2* expression by using the TCGA-LIHC (the Cancer Genome Atlas Liver Hepatocellular Carcinoma) cohort data as described above for the response to the same comment received from Reviewer 1, Comment #8 and Reviewer 3, Comment #8. To test the proposed *LTB4R2* role in human HCC, human HCC organoid data are included in **Fig. 4e** which clearly show tumor suppressive effects of *LTB4R2* KD or antagonism particularly with the organoid with higher *LTB4R2* expression.

3. “The authors should provide full-length WB for the images.”

Response: Due to the space constraint in figures, we have included all original full-length immunoblot images in the **Suppl Resource Folder**.

REVIEWERS' COMMENTS

Reviewer #1 (Remarks to the Author):

Most of my concerns are addressed though the mechanism by which CTNNB1-mediated regulation of LTB4R2 and CYP1B1 remains unclear. Also, Fig 4d. would have been better if the staining was done by multiplex IHC to show better that LTB4R2-expressing HCC cells are surrounded by CYP1B1+ CAFs. Markers of CAFs and HCC should also be stained together in the multiplex. And how many clinical samples were analyzed? Was only 1 clinical sample examined or was only 1 representative sample shown? This was not clear.

Reviewer #2 (Remarks to the Author):

The authors have addressed my concerns. I have no further concerns with this manuscript.

Reviewer #3 (Remarks to the Author):

The authors have performed a large amount of experiments and addressed most of my comments. There are two major points remaining (those can be addressed non-experimentally) and a few minor points that either require experimentation or further explanation.

Major points:

1. The title and abstract are not supported by data in the manuscript. They imply a tumor-promoting role of YAP1 that the authors have not demonstrated functionally. Moreover, from the authors' data, also showing changes in TAZ, and data from the literature showing an important role of TAZ in hepatocarcinogenesis, both YAP and TAZ may be part of this cascade. This needs to be altered or the authors need to show the functional involvement (there are several unpublished studies showing no major role for YAP in carcinogenesis in multiple organs as well as a study by Moya et al, *Science*. 2019;366(6468):1029-1034, showing tumor-restricting roles of YAP and TAZ).
2. A recent publication by Filliol et al, *Nature*. 2022;610(7931):356-365, shows that the main effect of stellate cells is outside tumors, and that stellate cells can promote and suppress tumors. These results are relevant for the current study, especially for where the proposed leukotriene B4 receptor2- β -catenin-YAP1 signaling cascade acts and promotes HCC, and should be discussed in the manuscript (HSCs are mostly outside tumors and most but not all of their tumor-promoting effects appear to be mediated outside tumors). The protective functions of HSCs in hepatocarcinogenesis should also be mentioned in the introduction.

Minor points:

1. The over 80% reduction of Scd2 mRNA in the liver in Fig.1g does not fit with the expression of Scd2 shown in Extended Data Fig.1a-b in other cell types, including endothelial cells (which are usually as abundant as HSCs, hence the reduction of 80% is odd; scRNA-seq, as in Ext.Data Fig.1a-b does not provide accurate representation of cell types as for example nuc-seq of whole liver). Please explain in the manuscript.
2. Fig.1e shows increased nuclear YAP outside tumors in floxed mice and a decrease in deleted mice. However, this is not quantified. Please address this point – as YAP outside tumors may be important.
3. The authors use the term “deletion in hepatocytes” for Alb-Cre. This should be corrected as Alb-Cre also deletes in cholangiocytes (stated in the initial review already).

4. The AAV8 data in hepatocytes in Fig.4a looks good and has quantification, but the tumor data in Fig.4b does not look convincing (much of the green fluorescence could be background and there are only 1-3 cells that are really green, one oddly shaped) and lacks a quantification. The data does not quite match what is shown in Ext.Data 4c. Please provide quantification – and if possible, also increase the n for 4c to make this more robust (it is an important point as AAV8 is given only 1 month before euthanizing mice).

Reviewer #4 (Remarks to the Author):

Authors have addressed all the concerns raised during initial review process. I recommend acceptance of this manuscript.

Point-by-point response to the reviewers' comments to the revised manuscript (NCOMMS-22-00840A)

Reviewer #1:

Comments:

Most of my concerns are addressed though the mechanism by which CTNNB1-mediated regulation of *LTB4R2* and *CYP1B1* remains unclear. Also, Fig 4d. would have been better if the staining was done by multiplex IHC to show better that *LTB4R2*-expressing HCC cells are surrounded by *CYP1B1*+ CAFs. Markers of CAFs and HCC should also be stained together in the multiplex. And how many clinical samples were analyzed? Was only 1 clinical sample examined or was only 1 representative sample shown? This was not clear.

Response:

Our future study will examine the mechanism of CTNNB1-mediated regulation of *LTB4R2* and *CYP1B1* by ChIP assay and intronic enhancer/promoter-reporter assays with or with site-directed mutagenesis of suspected CTNNB1 binding sites.

We have attempted multiplex staining for human HCC sections but unfortunately encountered major difficulties in identifying optimal antibodies for concomitant staining of *LTB4R2* with AFP or nuclear YAP1 for HCC cells and *CYP1B1* with FBLN2 or ACTA2 for CAFs. We would like to continue to search for optimal antibodies and staining conditions in future.

Fig 4d shown is one representative sample of at least four clinical samples analyzed, which we described in the revised figure legend.

Reviewer #2:

Comments:

The authors have addressed my concerns. I have no further concerns with this manuscript.

Response:

Thank you.

Reviewer #3:

Comments:

The authors have performed a large amount of experiments and addressed most of my comments. There are two major points remaining (those can be addressed non-experimentally) and a few minor points that either require experimentation or further explanation.

Major points:

1. The title and abstract are not supported by data in the manuscript. They imply a tumor-promoting role of YAP1 that the authors have not demonstrated functionally. Moreover, from the authors data, also showing changes in TAZ, and data from the literature showing an important role of TAZ in hepatocarcinogenesis, both YAP and TAZ may be part of this cascade. This needs to be altered or the authors need to show the functional involvement (there are several unpublished studies showing no major role for YAP in carcinogenesis in multiple organ as well as a study by Moya et al, Science. 2019;366(6468):1029-1034, showing tumor-restricting roles of YAP and TAZ).

Response:

Yes, our additional data suggest the potential involvement of TAZ in our proposed tumor-promoting cascade. However, we do not know how SCD in aHSC positively regulates TAZ and this new question will be addressed by our future study as done for YAP1 in the present study. For this reason, TAZ was not added to the title and abstract. To definitively confirm the functional role of YAP1 in liver tumorigenesis in our model, an additional direct loss-of function study for YAP1 is required in the model used, which we consider to be beyond the main scope of the study. The main thrust of the present study is the novel cascade initiated by a paracrine action of SCD by activated hepatic stellate cells, causing CTNNB1-YAP1 activation via *LTB4R2*

in tumor cells. The reviewer's point is correct that we still do not know the causal role of YAP1 activated by LTB4R2-CTNNB1 pathway in liver tumorigenesis. For this reason, we removed YAP1 from the title.

2. A recent publication by Filliol et al, Nature. 2022;610(7931):356-365, shows that the main effect of stellate cells is outside tumors, and that stellate cells can promote and suppress tumors. These results are relevant for the current study, especially for where the proposed leukotriene B4 receptor2- β -catenin-YAP1 signaling cascades acts and promotes HCC and should be discussed in the manuscript (HSC are mostly outside tumors and most but not all of their tumor-promoting effects appear to be mediated outside tumors). The protective functions of HSC in hepatocarcinogenesis should also be mentioned in the introduction.

Response:

Yes, this recent Nature publication elegantly reveals dual roles of HSC subpopulations mostly exerting their effects outside tumors. For this reason, we believe that our rationale for analyzing tumor-adjacent tissues without visible tumors was important and justified. We have added this discussion on HCC protective v. promoting roles of HSC subpopulations in the last paragraph of the revised Discussion.

Minor points:

1. The over 80% reduction of *Scd2* mRNA in the liver in Fig.1g does not fit with the expression of *Scd2* shown in Extended Data Fig.1a-b in other cell types, including endothelial cells (which are usually as abundant as HSCs, hence the reduction of 80% is odd; scRNA-seq, as in Ext.Data Fig.1a-b does not provide accurate representation of cell types as for example nuc-seq of whole liver). Please explain in the manuscript.

Response:

As discussed in the manuscript, *Scd2* is the CTNNB1-dependent gene expressed by HSCs and appears to have positive effects on *Scd1/2* expressed by other cell types via global CTNNB1 activation. For this reason, it is not surprising that ablation of HSC *Scd2* has a global suppressive effect. This discussion has been added in the 1st paragraph of page 5.

2. Fig.1e shows increased nuclear YAP outside tumors in floxed mice and a decrease in deleted mice. However this is not quantified. Please address this point – as YAP outside tumors may be important.

Response:

Data on nuclear YAP1+ cells outside tumors are shown in Fig. 1d as nYAP1+ HNF4A+ hepatocytes, ACTA2+ activated HSCs, and SOX9+ ductular cells.

3. The authors use the term “deletion in hepatocytes” for Alb-Cre. This should be corrected as Alb-Cre also deletes in cholangiocytes (stated in the initial review already)

Response:

We revised the manuscript to state, “lacking CTNNB1 primarily in hepatocytes but also in cholangiocytes” – 2nd paragraph of page 5.

4. The AAV8 data in hepatocytes in Fig.4a looks good and has quantification, but the tumor data in Fig.4b does not look convincing (much of the green fluorescence could be background and there are only 1-3 cells that are really green, one oddly shaped) and lacks a quantification. The data does not quite match what is shown in Ext.Data 4c. Please provide quantification – and if possible, also increase the n for 4c to make this more robust (it is an important point as AAV8 is given only 1 month before euthanizing mice).

Response:

We have noticed variable GFP expression in AFP+ tumor cells but the weaker GFP shown is not background. We now included a bar graph for the percentage of GFP+/AFP+ cells in the two groups – Supplementary (converted to Supplementary Fig from Ext. Data Fig per the editorial instructions) Fig. 4b. Data presented in Supplementary Fig 4c is sufficiently powered to show a statistically significant difference ($p=0.038$) in *Ltb4r2* mRNA level between the groups.

Reviewer #4

Comments:

Authors have addressed all the concerns raised during initial review process. I recommend acceptance of this manuscript.

Response:

Thank you.